# Towards Fine-Grained Robustness: Attention-Guided Test-Time Prompt Tuning for Vision-Language Models

Jia-Wei Hai [1]   Yijun Wang [1]   Xiu-Shen Wei [1 2]

## Abstract

Vision-Language Models (VLMs), such as CLIP, have achieved significant zero-shot performance on downstream tasks with various fine-tuning adaptation methods. However, recent studies have proven that adversarial attacks can significantly degrade the inference ability of VLMs, posing substantial risks to their practical applications. Prevalent test-time adaptation methods typically rely on multi-view augmentation to implement various fine-tuning strategies, which struggle to identify semantic information and are prone to destroying discriminative regions in fine-grained scenarios. To address these limitations, we propose Attention-Guided Test-Time Prompt Tuning (A-TPT), a semantics-preserving method designed for test-time adaptation. We first refine the gradient attention rollout mechanism to identify semantically meaningful regions surviving under adversarial attacks. Furthermore, we leverage them to guide the spatially varying augmentation intensities and multi-view ensemble for prompt tuning and inference. Extensive experiments demonstrate that A-TPT outperforms existing test-time adaptation methods on both adversarial and clean data. Codes are available at https://github.com/SEU-VIPGroup/A-TPT.

## 1. Introduction

Vision-Language Models (VLMs) pretrained on large-scale image–text pairs (Chen et al., 2023) have become a versatile foundation for downstream vision tasks (Shin et al., 2022; Zhang et al., 2023; Wei & Wei, 2024; Tianyuan et al.,

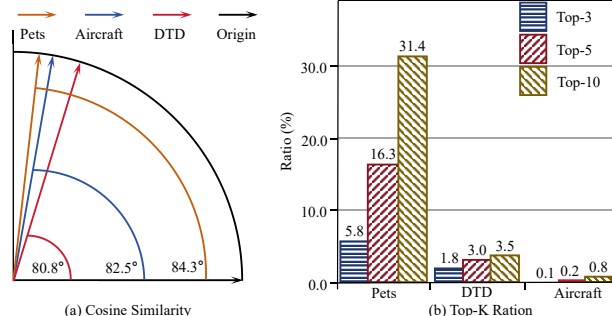

*Figure 1.* (a) Cosine similarity in the unit circle: adversarially attacked **(colored)** and original **(black)** feature vectors are highly divergent; (b) Ratio of true labels among the Top-K predictions under adversarial attacks: the true label of the input is pushed out of the Top-K predictions (ViT-B/16).

2026). However, they exhibit significant degradation under even subtle adversarial perturbations, raising serious safety concerns for real-world deployment (Zhao et al., 2023; Yu et al., 2026; Maolin et al., 2025). Training-time adaptation methods such as VPT (Li et al., 2024) and FAP (Zhou et al., 2024), among others (Hossain & Imteaj, 2025; Dong et al., 2025), have proven to be effective defense strategies. However, their reliance on large amounts of labeled adversarial data incurs high computational costs, limiting practicality. Although advanced test-time adaptation methods (Shu et al., 2022; Yoon et al., 2024; Xing et al., 2025) are primarily designed for natural distribution shifts, they provide limited robustness when the feature space is adversarially distorted.

Based on multiple embeddings derived from augmented views of a test sample (Hendrycks et al., 2021a; Feng et al., 2023), existing test-time defense methods (Tong et al., 2025; Wang et al., 2025a; Zanella & Ben Ayed, 2024) have proposed various optimization strategies to resist adversarial attacks on generic data. However, they often perform limited on fine-grained recognition, since random region-editing behavior leads to the loss of discriminative regions (Pu et al., 2024). Identifying and preserving discriminative semantic parts is the key to distinguishing fine-grained categories.

Existing semantics-preserving augmentation methods (Wang et al., 2025b; Ye et al., 2025; Tan et al., 2025) heavily rely on the quality of semantic directions and operate in the feature space, where attacked feature representations can

---

[1]School of Computer Science and Engineering, and Key Laboratory of New Generation Artificial Intelligence Technology and Its Interdisciplinary Applications, Southeast University, China. [2]School of Intelligence Science and Engineering, Southeast University, China. Correspondence to: Xiu-Shen Wei <weixs.gm@gmail.com>.

*Proceedings of the 43rd International Conference on Machine Learning*, Seoul, South Korea. PMLR 306, 2026. Copyright 2026 by the author(s).

be driven beyond the original decision boundary, as shown in Figure 1 (a). Among unsupervised semantics-preserving augmentation methods (Wang et al., 2019b; Pu et al., 2024; Ruru et al., 2024), the logits of an input are commonly used as the self-supervised signal, which loses label-preserving ability under adversarial attacks ( Figure 1 (b) ). Moreover, semantic identification is often implemented as a regularization term in the training loss, making it difficult to decouple from the training stage.

*In this paper, our central question is therefore: can we design a test-time adaptation method that identifies semantic information and leverages discriminative regions without learning in feature space even under adversarial attacks?* To this end, we propose Attention-Guided Test-Time Prompt Tuning (A-TPT), which utilizes robust visual attention as a semantic anchor to guide augmentation and multi-view ensemble. We first refine the Gradient Attention Rollout (GAR) (Chefer et al., 2021), since obtaining reliable semantic anchors requires robust attention maps. Although GAR is commonly used to identify discriminative regions, we demonstrate its vulnerability to adversarial perturbations. Specifically, we introduce a token-based gradient signal to refine its gradient computation, allowing attention to robustly highlight the discriminative regions against adversarial attacks. Following refined attention, A-TPT consists of two core modules: Attention-Guided Multi-View Augmentation and TV-Based Ensemble. Attention-guided Multi-view Augmentation utilizes refined attention to guide the spatially varying augmentation intensities at image-level, preserving high-attention regions while generating a set of diverse views. To further leverage semantically meaningful views, we introduce a TV-based ensemble to measure the reliability of each view and weight them. We hypothesize that views with large anisotropic Total Variation (TV) in their attention maps are semantically corrupted, either by adversarial attacks or irrelevant background. Thus, views with lower anisotropic TV are more reliable for containing discriminative semantic parts and contribute more to the ensemble. Finally, prompt tuning and the final inference build upon reliable views generated along the semantically meaningful directions. Extensive experiments tasks demonstrate that A-TPT not only identifies and preserves discriminative regions, but also substantially outperforms state-of-the-art (SOTA) test-time adaptation methods.

In summary, our main contributions are:

- We refine the gradient signal of GAR with a simple but robust term against adversarial perturbations, enabling effective identification of unperturbed semantic information under adversarial attacks.

- We propose A-TPT, the first method that exploits discriminative semantic regions for guiding test-time adaptation, particularly in fine-grained scenarios.

- Extensive experiments on nine datasets demonstrate that A-TPT substantially outperforms existing methods on both clean and adversarial data distributions.

## 2. Related Works

### 2.1. Adversarial Attacks and Defenses

Adversarial attacks on Vision-Language Models (VLMs) can significantly degrade model inference with even minimal perturbations (Szegedy et al., 2014). As a result, a variety of attack methods to generate adversarial samples have been proposed (Croce & Hein, 2020; Madry et al., 2018; Goodfellow et al., 2015). Common approaches like FGSM (Goodfellow et al., 2015) generate adversarial examples by perturbing in the direction of the loss gradient. PGD (Madry et al., 2018), a standard measure of model robustness, further projects gradient descent onto a perturbed $L_p$ ball with an $\varepsilon$-bounded constraint. Additionally, multi-modal adversarial attacks have emerged to jointly exploit weaknesses in both modalities. Co-Attack (Zhang et al., 2022) and VLATTACK (Yin et al., 2023) extend single-modal attacks by perturbing both the image and text encoders, thereby creating more effective adversarial examples.

Various defense methods have been proposed to counter adversarial attacks (Hossain & Imteaj, 2025; Zhou et al., 2024; Cui et al., 2024). Training on adversarial data improves the robustness of VLMs. SLADE (Hossain & Imteaj, 2025) aligns clean and adversarial embeddings at both patch and global scales via symmetric stop-gradient contrastive learning to enhance robustness. FAP (Zhou et al., 2024) enforces joint clean and adversarial semantic alignment through optimizing shared prompts, ensuring consistency between clean and adversarial distributions. Restoring adversarial inputs through a diffusion model is also a common but costly defense strategy (Nie et al., 2022; You et al., 2023). However, their training not only requires extra data costs but also compromises generalization across different datasets.

### 2.2. Test-time Adversarial Adaptation for VLMs

Test-time adaptation (Shu et al., 2022; Abdul Samadh et al., 2023; Sui et al., 2025) focuses on zero-shot generalization of pretrained models across test data, aiming to improve efficiency. Classic studies such as C-TPT (Yoon et al., 2024), DiffTPT (Feng et al., 2023), and VPT (Jia et al., 2022) have achieved significant performance on out-of-distribution (OOD) benchmarks. However, they overlook robustness under adversarial perturbations, which are more closely aligned with the quality of real-world data.

Recent studies (Zanella & Ben Ayed, 2024; Tong et al., 2025; Wang et al., 2025a) extend to adversarial situations and propose different strategies based on multi-view augmentation. MTA (Zanella & Ben Ayed, 2024) applies mean-

shift augmentation and optimizes feature representations through aggregation of multiple views, which focuses on the consistent mode and density peak in the feature space. AOM (Tong et al., 2025) introduces Gaussian noise into multiple embeddings and optimizes both the original and augmented anchor feature representations by minimizing Euclidean distance. Methods based on entropy optimization fine-tune model parameters or prompts. TAPT (Wang et al., 2025a) optimizes both textual and visual prompts using the marginal entropy across augmented views. The recent prompt tuning method R-TPT (Sheng et al., 2025) has demonstrated that entropy-based optimization yields strong robustness against adversarial attacks. R-TPT performs image-level augmentation and proposes pointwise entropy optimization for textual prompts across views. However, their augmentation mechanism suffers from destroying discriminative regions and semantically irrelevant background.

### 2.3. Semantics-Preserving Augmentations

Semantics-preserving augmentations (Wang et al., 2021; Hu et al., 2024; Yang et al., 2025) are typically employed in fine-grained scenarios, aiming to diversify samples along semantically meaningful directions. FN-NET (Ye et al., 2025) exchanges discriminative regions within the same subclass in the feature space and employs a distillation loss to facilitate the learning process. NAS (Wang et al., 2025b) learns a visual dictionary as the semantic bridge between visual and linguistic representations to generate augmented samples at the token level. Studies (Wang et al., 2019b; Pu et al., 2024) use original logits as the self-supervised label to train a learnable data aug-network, which is commonly used to preserve discriminative parts. Although these semantics-preserving augmentation methods are widely explored, they are rarely adopted for test-time adaptation because learning semantic information in the feature space is difficult to decouple from the training stage (as discussed in Appendix A). Methods (Michaeli & Fried, 2024; Islam et al., 2024; Li & Li, 2025) based on synthetic data can effectively generate high-quality views but are limited by costly diffusion models.

## 3. Methodology

We propose Attention-Guided Test-Time Prompt Tuning (A-TPT) to effectively identify and leverage discriminative semantic regions that remain under adversarial attacks. In Sec. 3.1, we review the foundational background of CLIP, test-time prompt tuning and gradient attention rollout. We then introduce three components of A-TPT in Sec. 3.2.

### 3.1. Preliminaries

**CLIP**   A widely used backbone in zero-shot inference topics, CLIP (Radford et al., 2021), has established a standard paradigm of contrastive learning for VLMs. Considering a

$C$-way classification task with class names $\{t_c\}_{c=1}^C$, CLIP extracts textual features $\mathbf{g}_c$ using a prompt template (*e.g.*, "a photo of a {class}") with the text encoder $G(\cdot)$, and visual features $\mathbf{f}_i$ with the image encoder $F(\cdot)$. The probability that a test sample $x_i$ belongs to class $c$ is given by:

$$p_c(x_i) = \frac{\exp\big(\cos(\mathbf{f}_i, \mathbf{g}_c)/\tau\big)}{\sum_{j=1}^C \exp\big(\cos(\mathbf{f}_i, \mathbf{g}_j)/\tau\big)}, \qquad (1)$$

where $\cos(\cdot, \cdot)$ denotes cosine similarity and $\tau$ is a temperature. The predicted label is $\hat{c} = \arg\max_c p_c(x_i)$.

**Test-Time Prompt Tuning (TPT)**   Methods like TPT (Shu et al., 2022), C-TPT (Yoon et al., 2024), and RTPT (Sheng et al., 2025) have focused on tuning textual prompts $P$ at test time by minimizing prediction entropy across various augmented views $\{x_i\}_{i=0}^N$ of a test sample $x_0$. The optimization of TPT without additional training priors is:

$$\mathcal{L}_H(p) = -\frac{1}{|\mathcal{B}|} \sum_{i \in \mathcal{B}} \sum_{c=1}^C p_c(x_i) \log p_c(x_i), \qquad (2)$$

where $\mathcal{B}$ denotes a set of low-entropy samples selected from all augmented views $\{x_i\}_{i=0}^N$. Intuitively, $P$ is updated for $T$ steps by gradient descent to refine the decision boundary:

$$P^{(T)} = P^{(T-1)} - \eta \nabla_P \mathcal{L}_H(P^{(T-1)}). \qquad (3)$$

**Gradient Attention Rollout (GAR)**   GAR (Chefer et al., 2021; Gupta et al., 2022) element-wise weights attention matrices using the gradient of a target logit, and obtains the final CLS-to-patch attribution after cross-layer rollout multiplication. Following GAR, attention focuses on the discriminative semantic parts for a specific target. For the $b$-th transformer block, let $s$ and $h$ denote the number of tokens and the number of heads in each attention layer, and attention tensor is denoted as $\mathbf{A}^{(b)}(x) \in \mathbb{R}^{h \times s \times s}$. Given the target logit $S(x) = \text{logits}_t(x)$ for an input $x$, GAR constructs a gradient-weighted transition matrix $\hat{\mathbf{A}}^{(b)}(x)$ at the $b$-th block as follows:

$$\hat{\mathbf{A}}^{(b)}(x) = \mathbf{I} + E_h\Big(\nabla_{\mathbf{A}^{(b)}(x)} S(x) \odot \mathbf{A}^{(b)}(x)\Big)^+, \quad (4)$$

where $\odot$ denotes the Hadamard product, $\nabla_{\mathbf{A}^{(b)}(x)} S(x)$ is the gradient matrix corresponding to the attention matrix $\mathbf{A}^{(b)}(x)$, and $\mathbf{I}$ is the identity matrix accounting for residual connections. $E_h(\cdot)^+$ denotes the mean operation over the head dimension, with negative values clamped to zero. The final transition matrix $\hat{\mathbf{A}}(x)$ is obtained by recursively multiplying over $B$ blocks:

$$\hat{\mathbf{A}}(x) = \prod_{b=1}^B \hat{\mathbf{A}}^{(b)}(x) \in \mathbb{R}^{s \times s}. \qquad (5)$$

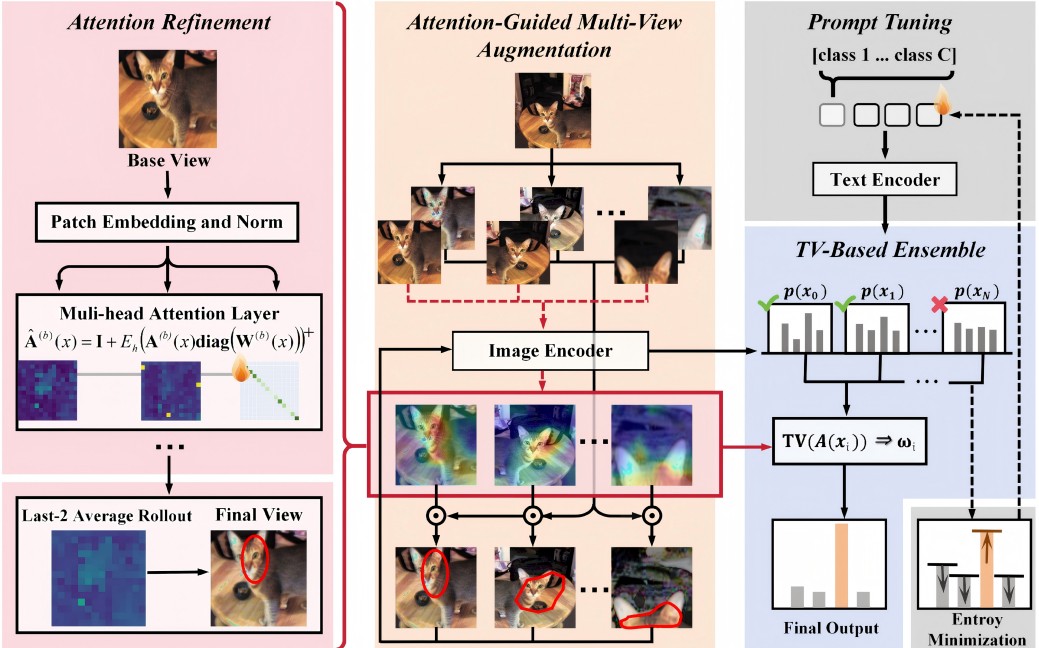

*Figure 2.* The pipeline of A-TPT. Given an input sample, Attention Refinement based on token-gradient is used to identify semantic parts. Then, Attention-Guided Multi-View Augmentation builds a set of semantics-preserving views for fine-tuning learnable prompts. After selecting low-entropy views followed by prompt tuning, TV-Based Ensemble weights reliable views in the final inference process.

GAR takes the first row of $\hat{\mathbf{A}}(x)$ corresponding to the $[CLS]$ token and removes the attention of the $[CLS]$ token with itself. The resulting attention embedding of size $(s - 1)$ is reshaped to size $(\sqrt{s-1} \times \sqrt{s-1})$ to obtain the final CLS-to-patch attention map $\mathbf{A}(x)$.

### 3.2. The A-TPT Framework

A-TPT performs semantics-preserving prompt tuning in attention space and requires neither training knowledge nor additional models. As depicted in Figure 2, *1) Attention Refinement* first accurately identifies unperturbed fine-grained parts in specific multiple views; *2) Attention-Guided Multi-View Augmentation* is responsible for semantics-preserving augmentation; *3) TV-Based Ensemble* weights the multiple views based on the quality of attention distribution, filtering the semantic-empty and semantic identifying failed views.

**Gradient Attention Refinement** Our attention-guided multi-view augmentation described in Eq. (9)–(11) requires a robust semantic anchor $\mathbf{A}(x) \in \mathbb{R}^{H \times W}$. Existing gradient signal $\nabla_{\mathbf{A}^{(b)}(x)} S(x)$ used in GAR is sensitive to adversarial perturbations, introducing scattered isolated peaks in the spatial attention map as proven in Appendix B.

Therefore, we replace the attention-based gradient signal with a robust token-based gradient signal as the target-specific weighting. Specifically, let $\mathbf{T}^{(b)}(x) \in \mathbb{R}^{s \times d}$ denote the token embeddings entering the $b$-th block. We define a token-based weighting vector as following:

$$\mathbf{W}^{(b)}(x) = \mathcal{N}\left( \left[ \langle \mathbf{T}^{(b)}(x), \nabla_{\mathbf{T}^{(b)}(x)} S(x) \rangle_d \right]_+ \right), \quad (6)$$

where $\langle \cdot, \cdot \rangle$ is the inner product over the embedding dimension $d$, $[\cdot]_+$ clamps negatives to 0, $\mathcal{N}(\cdot)$ denotes $\ell_1$ normalization. We then weight the attention matrix along the source-token dimension via column scaling and our gradient-weighted transition matrix at block $b$ is:

$$\hat{\mathbf{A}}^{(b)}(x) = \mathbf{I} + E_h \left( \mathbf{A}^{(b)}(x) \operatorname{diag}\left( \mathbf{W}^{(b)}(x) \right) \right)^+. \quad (7)$$

Additionally, we retain only the last two transition matrices $\hat{\mathbf{A}}^{(B-1)}(x)$ and $\hat{\mathbf{A}}^{(B)}(x)$ to suppress shallow-layer noise and apply a last-two-layer average stabilization. The average transition matrix of last two layers is:

$$\hat{\mathbf{A}}_{\mathrm{avg}}(x) = \frac{\hat{\mathbf{A}}^{(B-1)}(x) + \hat{\mathbf{A}}^{(B)}(x)}{2}. \quad (8)$$

Finally, $\hat{\mathbf{A}}(x) = \hat{\mathbf{A}}^{(B)}(x) \hat{\mathbf{A}}_{\mathrm{avg}}(x)$ is used to obtain the final CLS-to-patch attention $\mathbf{A}(x)$. In summary, we replace the weighting term $\mathbf{A}^{(b)}(x) \odot \nabla_{\mathbf{A}^{(b)}(x)} S(x)$ with $\mathbf{A}^{(b)}(x) \operatorname{diag}(\mathbf{W}^{(b)}(x))$ to decouple the weighting from attention-based gradient signal. This substitution avoids the second-order sensitivity of $\nabla_{\mathbf{A}^{(b)}(x)} S(x)$ to perturbations, which can induce scattered peaks in the final attention map and destabilize subsequent mask construction described in Eq. (9)–(11). By contrast, $\mathbf{W}^{(b)}(x)$ aggregates contributions over the embedding dimension via source-token

column scaling rather than operating on individual attention edges, and the resulting CLS-to-patch attribution is smoother and reliable for masks. (Proof in Appendix B).

**Attention-Guided Multi-View Augmentation** Multi-view augmentation helps mitigate adversarial noise and improve prediction consistency. However, for fine-grained tasks, noisy augmentations often corrupt discriminative parts (Tan et al., 2025). To address this issue, we propose attention-guided multi-view augmentation. We treat the robust attention map, described in Eq. (6)–(8), as a semantic anchor and apply spatially varying augmentation intensities: high-attention regions are prioritized for semantic protection, while low-attention regions emphasize diversity.

Given a test sample $x_0$, we first construct a set of base views $\{b_i\}_{i=0}^N$ using Random-Flip and Center-Crop, along with aggressive views $\{\tilde{x}_i\}_{i=0}^N$ generated by AugMix. The spatial attention $\mathbf{A}(b_i) \in \mathbb{R}^{H \times W}$ of a base view is extracted from the visual encoder. We divide spatial locations into high- and low-attention regions using a ratio $r$. Let $J = \lceil rHW \rceil$, and $A_{(J)}(b_i)$ denote the $J$-th largest value of $\mathbf{A}(b_i)$ (the top-$r$ threshold). For any spatial location with attention value $A$, we define the high-attention mask as:

$$M_{\text{high}}(r) = \begin{cases} 1, & A \geq A_{(J)}(b_i), \\ 0, & A < A_{(J)}(b_i), \end{cases} \qquad (9)$$

where the low-attention mask is $M_{\text{low}}(r) = 1 - M_{\text{high}}(r)$. The Mixing strength $\lambda(r) \in \{m_{\text{high}}, m_{\text{low}}\}$ in the high-attention and low-attention regions is defined as:

$$\lambda(r) = M_{\text{high}}(r)\, m_{\text{high}} + M_{\text{low}}(r)\, m_{\text{low}}, \qquad (10)$$

then, attention-guided augmented views are generated as:

$$x_i = \big(1 - \lambda(r)\big) \odot b_i + \lambda(r) \odot \tilde{x}_i. \qquad (11)$$

**TV-Based Ensemble of Multiple Views** The TV-based ensemble strategy is employed to weight predictions from selected low-entropy views and obtain a reliable final output. Specifically, for each selected view $\tilde{x}_i$ and its attention map $\mathbf{A}(x_i) \in \mathbb{R}^{H \times W}$, we quantify the spatial fragmentation of an attention map by the anisotropic Total Variation (TV):

$$\text{TV}(\mathbf{A}(x_i)) = \sum_{u=1}^{H-1} \sum_{v=1}^{W} |A_{u+1,v} - A_{u,v}| \\ + \sum_{u=1}^{H} \sum_{v=1}^{W-1} |A_{u,v+1} - A_{u,v}|. \qquad (12)$$

Here $A_{u,v}$ is the value at location $(u, v)$ in the attention map of each view. This expression sums value differences between neighboring patches along vertical and horizontal directions. $|A_{u+1,v} - A_{u,v}|$ and $|A_{u,v+1} - A_{u,v}|$ measure

the magnitude of attention change between neighboring patches. The first term compares two vertically adjacent positions $(u, v)$ and $(u+1, v)$. The second term compares two horizontally adjacent positions $(u, v)$ and $(u, v+1)$. Large TV indicates that $A_{u,v}$ varies drastically, which means more fragmented attention maps with stronger local abrupt changes. In contrast, small TV means smoother and more continuous with more consistent neighboring position in attention maps. Our experiments demonstrate that more fragmented or overly peaky attention distributions often indicate invalid augmented views, such as background-dominated views or views corrupted by high-contrast artifacts. Therefore, we use TV to measure the reliability of each view and obtain weights $w_i$ for the final prediction $\hat{c}$:

$$w_i = \frac{\exp\big(-\text{TV}(\mathbf{A}(x_i))\big)}{\sum_{j \in \mathcal{B}} \exp\big(-\text{TV}(\mathbf{A}(x_j))\big)}, \qquad (13)$$

$$\hat{c} = \arg\max_c \sum_{i \in \mathcal{B}} w_i\, p_c(x_i). \qquad (14)$$

## 4. Experiments

To comprehensively evaluate the effectiveness of A-TPT, our experimental results are organized into three sections. In Sec. 4.2, we benchmark A-TPT against advanced test-time defense methods on both clean and adversarial data. Sec. 4.3 presents extended experiments, comparing A-TPT with training-time adaptation methods and evaluating its robustness under more attacks. Finally, Sec. 4.4 conducts ablation studies to justify the motivation of our framework and quantify the contribution of each component.

### 4.1. Setups

**Datasets** Previous works (Wang et al., 2024; Xing et al., 2025) mainly focus on adversarial generalization on ImageNet-1K (Deng et al., 2009) and its variants, but often ignore fine-grained scenarios. In contrast, our method effectively leverages discriminative semantic parts of the fine-grained targets while improving robustness. To evaluate the effectiveness, especially for fine-grained tasks, we cover diverse classification tasks including Caltech101 (Fei-Fei, 2004), OxfordPets (Parkhi et al., 2012), Flower102 (Nilsback & Zisserman, 2008), StanfordCars (Krause et al., 2013), FGVC-Aircraft (Maji et al., 2013), DTD (Cimpoi et al., 2014), EuroSAT (Helber et al., 2019), and UCF101 (Soomro et al., 2012). We also conduct experiments on ImageNet to identify general ability. In addition, we also conduct experiments on ImageNet-out-of-distribution (OOD) datasets, including ImageNet-A (Hendrycks et al., 2021b), ImageNet-V2 (Recht et al., 2019), ImageNet-R (Hendrycks et al., 2021a), ImageNet-S (Wang et al., 2019a)

*Table 1.* Adversarial accuracy of test-time adaptation methods on ImageNet and fine-grained classification tasks via pretrained ViT-B/16 and ViT-B/32, with the best results shown in **bold red** (ViT-B/16) and **bold blue** (ViT-B/32).

| Methods | | Pets | Caltech101 | Cars | DTD | UCF101 | EuroSAT | Flower102 | Aircraft | Average |
|---|---|---|---|---|---|---|---|---|---|---|
| CLIP | ViT-B/16 | 0.0 | 0.0 | 0.0 | 0.0 | 0.0 | 0.0 | 0.0 | 0.0 | 0.0 |
| | ViT-B/32 | 0.2 | 0.0 | 0.0 | 0.0 | 0.0 | 0.0 | 0.0 | 0.0 | 0.0 |
| TTC | ViT-B/16 | 10.4 | 8.4 | 2.9 | 4.5 | 1.6 | 0.4 | 7.4 | 0.5 | 4.5 |
| | ViT-B/32 | 11.8 | 22.7 | 2.3 | 4.7 | 6.1 | 3.0 | 3.2 | 1.0 | 6.9 |
| TPT-Ensemble | ViT-B/16 | 51.2 | 74.7 | 26.0 | 25.1 | 30.6 | 2.2 | 36.3 | 8.7 | 31.9 |
| | ViT-B/32 | 52.5 | 74.9 | 25.9 | 28.6 | 36.9 | 11.9 | 36.1 | 7.9 | 34.3 |
| MTA | ViT-B/16 | 51.8 | 72.1 | 18.5 | 16.2 | 27.5 | 1.2 | 27.9 | 4.3 | 27.4 |
| | ViT-B/32 | 53.6 | 76.3 | 26.4 | 28.8 | 39.1 | 11.3 | 36.5 | 8.2 | 35.0 |
| R-TPT | ViT-B/16 | 60.2 | 82.0 | 34.7 | 32.8 | 43.2 | 8.5 | 44.6 | 13.2 | 39.9 |
| | ViT-B/32 | 55.8 | 76.4 | 28.4 | 29.1 | 41.0 | 5.1 | 37.6 | 9.2 | 35.3 |
| **A-TPT (Ours)** | ViT-B/16 | **70.5** | **85.6** | **39.2** | **37.8** | **51.7** | **13.1** | **52.6** | **15.1** | **45.7** |
| | ViT-B/32 | **66.4** | **79.8** | **31.8** | **31.1** | **46.9** | **12.7** | **43.2** | **10.4** | **40.3** |

*Table 2.* Clean accuracy of test-time adaptation methods on ImageNet and fine-grained classification tasks via pre-trained ViT-B/16 and ViT-B/32, with the best results shown in **bold red** (ViT-B/16) and **bold blue** (ViT-B/32).

| Methods | | Pets | Caltech101 | Cars | DTD | UCF101 | EuroSAT | Flower102 | Aircraft | Average |
|---|---|---|---|---|---|---|---|---|---|---|
| CLIP | ViT-B/16 | 88.3 | 94.0 | 65.5 | 44.4 | 65.2 | 42.2 | 67.4 | 23.9 | 61.4 |
| | ViT-B/32 | 85.1 | 91.4 | 60.1 | 43.0 | 61.6 | 35.8 | 64.0 | 18.1 | 57.4 |
| TTC | ViT-B/16 | 82.3 | 87.6 | 55.0 | 41.0 | 65.8 | **47.4** | 69.0 | 23.3 | 58.9 |
| | ViT-B/32 | 83.5 | 86.5 | 48.1 | 37.3 | 62.6 | **53.0** | 64.3 | 18.2 | 56.7 |
| TPT-Ensemble | ViT-B/16 | 86.2 | 91.9 | 65.7 | 43.2 | 63.0 | 28.2 | 65.9 | 23.4 | 58.4 |
| | ViT-B/32 | 75.0 | 88.2 | 51.7 | 39.8 | 54.9 | 30.8 | 58.1 | 16.4 | 51.9 |
| MTA | ViT-B/16 | 88.0 | 94.3 | 67.7 | 46.5 | 67.5 | 42.5 | 67.4 | **25.0** | 62.4 |
| | ViT-B/32 | 86.3 | 92.0 | 63.4 | 43.8 | 63.3 | 34.6 | 64.4 | **20.2** | 58.5 |
| R-TPT | ViT-B/16 | 87.2 | 93.7 | 67.0 | 46.4 | 67.2 | 34.7 | 68.7 | 23.9 | 61.1 |
| | ViT-B/32 | 84.5 | 90.6 | 63.1 | 42.1 | 62.8 | 32.0 | 62.6 | 19.1 | 57.1 |
| **A-TPT (Ours)** | ViT-B/16 | **89.5** | **94.6** | **67.9** | **48.0** | **68.9** | 42.2 | **70.1** | 22.5 | **63.0** |
| | ViT-B/32 | **87.3** | **92.0** | **63.8** | **44.0** | **63.4** | 35.9 | **64.5** | 17.5 | **58.6** |

**Baselines** We compare A-TPT against two test-time prompt tuning methods for CLIP: TPT-Ensemble (Shu et al., 2022) and the SOTA method R-TPT (Sheng et al., 2025), as well as a strong method based on the test-time distribution alignment for CLIP: MTA (Zanella & Ben Ayed, 2024). An advanced detection-then-defense method, TTC (Xing et al., 2025), is also included. Here, TPT-Ensemble employs simple averaging of predictions across selected low-entropy views. All baseline methods rely on pretrained CLIP and AugMix (Hendrycks et al., 2020) augmentation, without any other training knowledge or additional models.

**Evaluation Metrics** Following previous works (Sheng et al., 2025; Zanella & Ben Ayed, 2024), we report two average accuracies (%): clean accuracy is used to measure the method's adaptation ability on clean samples, while adversarial accuracy measures the method's robustness on adversarial samples.

**Implementation Details** We use the official pretrained CLIP models (ViT-B/16, ViT-B/32 and ResNet50) as the backbone throughout our experiments. PGD (Madry et al., 2018) is used to generate adversarial samples. Specifically, we use a perturbation bound of $\varepsilon = 4/255$ with 100 steps for ViT, and $\varepsilon = 1/255$ with 1 step for ResNet. In the prompt tuning stage at test time, we utilize the Adam optimizer with weight decay and one step with a learning rate of 0.005. All experiments are conducted using the PyTorch framework on RTX-4090 GPUs with eight-card data parallelism. More setup details are given in Appendix C.

### 4.2. Main Results

**Results of Adversarial Robustness** As shown in Table 1, we evaluate the adversarial robustness of A-TPT on eight fine-grained tasks and the ImageNet task. A-TPT achieves the best adversarial accuracy of $45.7\%$ (ViT-B/16) and $40.3\%$ (ViT-B/32), significantly outperforming existing test-time defense methods. This robustness highlights the broad applicability of A-TPT and its potential to serve as an effective defense paradigm for future vision-language models.

MTA aims to obtain consistent representations across aug-

*Table 3.* Clean and adversarial (Adv.) accuracy of test-time adaptation methods on ImageNet-OOD datasets (ResNet50).

| Methods | ImageNet | | ImageNet-A | | ImageNet-V2 | | ImageNet-R | | ImageNet-S | | Average | |
|---|---|---|---|---|---|---|---|---|---|---|---|---|
| | Clean | Adv. | Clean | Adv. | Clean | Adv. | Clean | Adv. | Clean | Adv. | Clean | Adv. |
| CLIP | 58.2 | 0.1 | 21.8 | 0.0 | 51.5 | 0.1 | 56.1 | 0.8 | 33.3 | 0.5 | 44.2 | 0.3 |
| TPT-Ensemble | 58.0 | 40.1 | 22.6 | 10.1 | 52.0 | 37.2 | 51.3 | 39.3 | 29.5 | 20.7 | 42.7 | 29.5 |
| MTA | 60.4 | 30.0 | 27.5 | 5.6 | 54.2 | 24.6 | **58.4** | 29.8 | **35.2** | 11.3 | 47.1 | 20.3 |
| R-TPT | 60.9 | 47.7 | 28.4 | 14.4 | 54.9 | 41.6 | 57.6 | 46.9 | 34.0 | 26.2 | 47.1 | 35.4 |
| **A-TPT (Ours)** | **61.4** | **48.2** | **31.6** | **15.3** | **55.1** | 42.1 | 57.6 | **47.1** | 34.4 | **26.4** | **48.0** | **35.8** |

mented views through aggregation optimization. RTPT introduces pointwise entropy of augmented views to optimize textual prompts. However, these methods overlook the semantic information. A-TPT can identify unperturbed discriminative parts and achieves semantics-preserving diversity, ultimately surpassing MTA by $5.3\%$ (ViT-B/32) and R-TPT by $5.0\%$ (ViT-B/32) in adversarial accuracy. Semantic preservation does matter, and what matters more is identifying discriminative semantic parts.

**Results of Clean Accuracy** Besides adversarial robustness, A-TPT also improves the clean accuracy of CLIP from $61.4\%$ to $63.0\%$ (ViT-B/16) and from $57.4\%$ to $58.6\%$ (ViT-B/32) as shown in Table 2. Compared to previous methods, we achieve the best performance and surpass the SOTA defense method RTPT by $1.5\%$ via ViT-B/32 in the clean distribution. These methods largely preserve CLIP's zero-shot ability, indicating that extracting consistent information from multiple views is a reliable approach. A-TPT can further leverage discriminative semantics to guide this process, whether or not the features are corrupted.

**Results on ImageNet OOD Datasets** Beyond the fine-grained scenarios, we further report the experimental results on five generic benchmarks in Table 3. Although CLIP can handle natural distribution shifts, it is highly vulnerable to adversarial perturbations. A-TPT achieves the strongest adversarial robustness across both ImageNet and its out-of-distribution variants. Notably, A-TPT reaches an average adversarial robustness of $35.8\%$, while CLIP achieves only $0.3\%$. In addition, A-TPT maintains clean accuracy comparable to baselines, suggesting our effectiveness on large-scale datasets with distribution shifts.

## 4.3. More Experiments

**Results under Different Adversarial Attacks** We evaluate A-TPT with two SOTA methods, MTA and R-TPT, under various attack methods. We employ CW (Carlini & Wagner, 2017), DeepFool (DF) (Moosavi-Dezfooli et al., 2016), and FGSM (Goodfellow et al., 2015) as new attacks and report the adversarial accuracy on two fine-grained datasets in Table 4. A-TPT achieves the highest adversarial accuracy across all types of attacks. Obviously, the attention-based

*Table 4.* Adversarial accuracies under various attacks on two fine-grained datasets (ViT-B/32).

| Methods | Flower | | | | DTD | | | |
|---|---|---|---|---|---|---|---|---|
| | CW | DF | FGSM | Avg. | CW | DF | FGSM | Avg. |
| MTA | 34.5 | 35.4 | 36.6 | 35.5 | 23.6 | 23.5 | 23.9 | 23.7 |
| R-TPT | 51.6 | 54.7 | 49.2 | 51.8 | 34.2 | 35.9 | 32.5 | 34.2 |
| **A-TPT (Ours)** | **54.7** | **58.2** | **53.4** | **55.4** | **39.2** | **41.6** | **38.8** | **39.9** |

*Table 5.* The average accuracy across 8 fine-grained datasets compared A-TPT with training-time defense methods (ViT-B/32).

| Methods | Stage | Clean Acc. | Adv. Acc. |
|---|---|---|---|
| PAFT | Training time | 54.9 | 27.5 |
| TeCoA | Training time | 33.4 | 32.2 |
| APT+TeCoA | Training time | 40.2 | 38.9 |
| FARE | Training time | 42.2 | 40.3 |
| **A-TPT (Ours)** | Test time | **58.6** | **40.3** |

semantics-preserving strategy of A-TPT is stable and substantial, confirming the simple but effective direction.

**Results Compared with Training-Time Methods** Training-time defense methods typically require labeled data and compromise performance on clean data. To further examine the effectiveness of A-TPT, we compared A-TPT with training-time adaptation methods PAFT (Chen et al., 2020), TeCoA (Mao et al., 2023), APT (Li et al., 2024), and FARE (Schlarmann et al., 2024) on fine-grained datasets, as shown in Table 5 (Table 9 and Table 10 in Appendix C). It is shown that A-TPT not only maintains the best performance on clean data but also achieves the best robustness on adversarial data. It confirms that the discriminative regions preserved by A-TPT's token-gradient attention cover most of the discriminative parts.

**Inference Efficiency** We further study the inference efficiency, finding that the additional computational cost stems from per-sample attention extraction, as shown in Table 6. Moreover, the inference can be accelerated by reducing the number of views to obtain better real-time performance. By reducing this number from 64 to 16, the time required for per-sample is reduced to less than 1/5 of the original, but the accuracy remains stable.

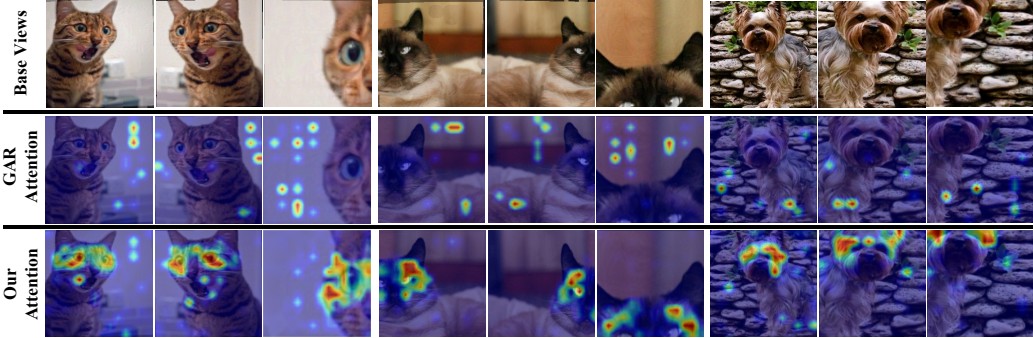

*Figure 3.* Quality of semantic identification on adversarial examples (Pets dataset). Compared with GAR, our refined attention focuses on continuous and discriminative semantic parts (ViT-B/16).

*Table 6.* Inference efficiency on the Pets dataset (ViT-B/16).

| Views Number | Time/per-sample | Adv. Accuracy |
|---|---|---|
| 64 | 2.71 s | 70.5 |
| 32 | 1.07 s | 70.1 |
| 16 | 0.53 s | 69.8 |

*Table 7.* Ablation studies: adversarial and clean accuracy on eight fine-grained datasets (ViT-B/16).

| A-Refine | A-Aug | A-TV | Adversarial Acc. | Clean Acc. |
|---|---|---|---|---|
| × | × | × | 31.8 | 58.4 |
| ✓ | × | × | 31.8 | 58.4 |
| × | ✓ | ✓ | 32.3 | 61.3 |
| ✓ | × | ✓ | 38.1 | 62.4 |
| ✓ | ✓ | × | 41.6 | 62.7 |
| ✓ | ✓ | ✓ | 45.7 | 63.0 |

## 4.4. Ablation Studies

**Quality of Semantic Identification**   The key to semantic preservation lies in whether a model can still accurately identify the discriminative parts under adversarial attacks. Figure 3 presents the base views and corresponding attention maps under adversarial attacks. GAR fails to locate the critical regions due to the affected term $\nabla_{\mathbf{A}^{(b)}(x)} S(x)$ in Eq. (4). In contrast, our refined attention can focus on the same part across different views generated from the same test sample, indicating that unperturbed semantic information is effectively captured under adversarial attacks.

**Reliability of TV-Based Ensemble**   To further demonstrate the reliability of TV-based ensemble, we visualize attention distribution across views assigned different weights, as shown in Figure 4. It can be seen that during the ensemble stage of A-TPT, the attention of highly reliable ($w_i > 0.01$) views is more evenly concentrated on discriminative parts. In contrast, the attention of low-reliability ($w_i < 0.001$) views is more dispersed, and these views often correspond to semantically irrelevant backgrounds or perturbed features. Obviously, TV-based ensemble effectively distinguishes between semantically meaningful and semantically empty views, assigning them different weights.

**Sensitivity of the Numbers of View**   We study the impact of the number of augmented views to further demonstrate the effectiveness of A-TPT. We provide the results of A-TPT with different view numbers (16, 32, 64) in Figure 5. The results show that A-TPT remains stable across different view numbers, indicating that with only a few views, A-TPT can accurately capture semantic information and effectively

guide multi-view augmentation.

**Contribution of Core Components**   We evaluate the contribution of each component included in A-TPT (A-Refine = Attention Refinement, A-Aug = Attention-Guided Multi-View Augmentation, and A-TV = TV-Based Ensemble), as shown in Table 7. The baseline is TPT-Ensemble, which conducts average ensemble after augmentation and prompt tuning. A-Refine is the foundation of A-TPT. Comparing the results in the second and third rows, it is evident that without A-Refine under adversarial attacks, A-Aug fails to protect discriminative regions and A-TV fails to ensemble reliable views. A-Aug and A-TV can effectively improve performance on both clean and adversarial data. Based on A-Refine, A-TV improves adversarial accuracy and clean accuracy by 6.3% and 4.0%, respectively, while A-Aug improves by 9.8% and 4.3%. This result further supports the conclusions in Sec. 4.3 regarding the Reliability of TV-Based Ensemble and Quality of Semantic Identification. Furthermore, applying A-Aug for discriminative parts protection, followed by A-TV to filter out views without useful semantic information, improves the accuracy to 45.7% and 63.0%, respectively. In summary, A-Aug and A-TV can fully leverage semantic information to guide prompt tuning when discriminative parts are located by A-Refine.

## 5. Conclusions and limitations

In this paper, we explored semantics-preserving adversarial defense for VLMs at test time and proposed Attention-

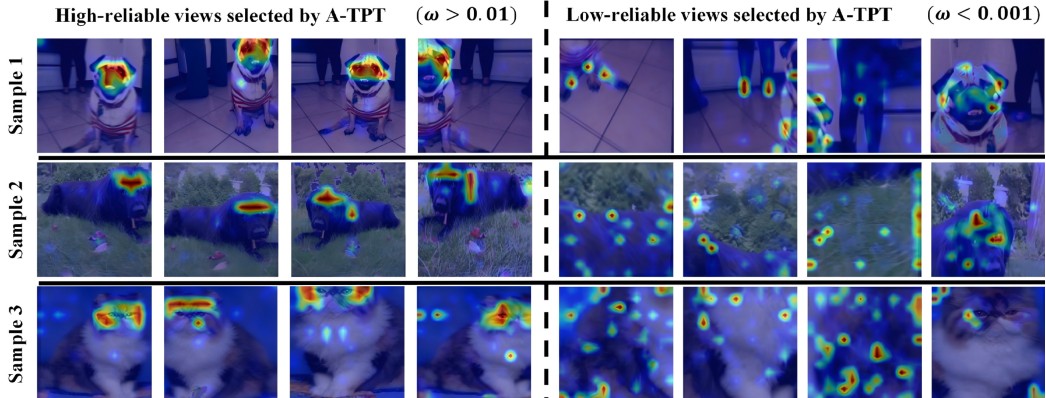

High-reliable views selected by A-TPT    ($\omega > 0.01$)    Low-reliable views selected by A-TPT    ($\omega < 0.001$)

Sample 1    Sample 2    Sample 3

*Figure 4.* Attention distribution of high-reliable views and low-reliable views from the same test sample on the Pets dataset (ViT-B/16).

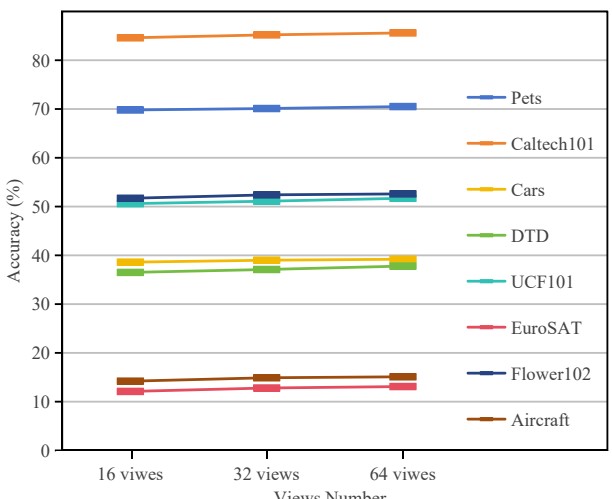

*Figure 5.* Adversarial accuracy with the various numbers of augmented views (ViT-B/16).

Guided Test-Time Prompt Tuning (A-TPT). Inspired by feature corruption, we first decoupled semantic identification from the training stage and leveraged the unperturbed semantic information under adversarial attacks. We found that existing gradient attention is sensitive to adversarial attacks and can introduce random attention distribution; thus, we replaced its gradient signal with a robust term to refine the attention strategy. This refined attention can guide the model to tune prompts across multiple views in a semantics-preserving way and ensemble views without corrupted features or useless backgrounds. Extensive results demonstrated that A-TPT achieves outstanding defense performance against various adversarial attacks and maintains adaptation performance on clean data. Our proposed method is built upon the views generated by multi-view transformations that preserve stable semantic structures. Consequently, A-TPT relies on multi-view augmentation, as well as view selection and ensemble, and is not entirely immune to adversarial attacks. In future work, we will further explore discriminative semantic information in the latent space.

## Acknowledgments

This work was supported by National Natural Science Foundation of China under Grant (62522602, 62272231), Basic Research Program of Jiangsu under Grant (BK20250073), the Fundamental Research Funds for the Central Universities (4009002401, 2242025K30024), and the Big Data Computing Center of Southeast University.

## Impact Statement

This paper presents work whose goal is to advance the field of Machine Learning. There are many potential societal consequences of our work, none which we feel must be specifically highlighted here.

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

# A. Discussion

## A.1 Why attention-guided augmentation is necessary in test-time prompt tuning?

Existing test-time prompt tuning methods (Wang et al., 2025a; Sheng et al., 2025; Shu et al., 2022; Yoon et al., 2024) use multi-view augmentation to rewrite the optimization objective as an unsupervised expectation. They minimize entropy/uncertainty or enforce cross-view consistency, yielding more stable gradients that are less prone to bias from distribution shift or adversarial perturbations.

The goal of attention-guided augmentation is to generate multiple views that preserve discriminative regions under adversarial perturbations. More specifically:

- Random augmentations can easily introduce spurious background or texture noises, which will corrupt fine-grained discriminative parts.

- Attention, as a semantic anchor, can guide augmentation to preserve high-semantic regions while still introduce diversity. Preserving target parts or discriminative regions can effectively reduce the number of semantic-empty views.

- Attention-guided augmentation improves the stability of optimization by less low-quality views, and the effectiveness of ensemble by more high-quality views.

## A.2 Why is the semantic anchor attention instead of feature representations?

The key issue is not simply " Where semantic information are? ", but " Is there still unperturbed and discriminative semantics exist? ". Although semantics-preserving augmentation has been widely explored (Wang et al., 2021; Hu et al., 2024; Yang et al., 2025), existing problems in test-time adversarial satiation are:

- Adversarial attacks will push the features away from the decision boundary, which break the structure of the similarity or semantic neighborhood in feature space. In such case, using attacked feature representations to identify discriminative semantics is adding fuel to the fire.

- Attention is a training-independent signal. Directly learning semantics in feature space requires pre-learned prototypes, clustering, or dictionaries. As the regularization terms, it is difficult to decouple them from the training stage.

## A.3 Why does A-TPT need TV-based ensemble?

Not all views retain discriminative semantics under adversarial attacks. Since multiple views are generated from adversarially perturbed samples, some of them inherit aggressive adversarial noise, causing target parts to be broken or mixed with the background. Therefore, we need a way to filter out useless views. When a view has no clear discriminative regions, its attention map exhibits larger total variation. So the TV-based ensemble serves as the last line of defense.

# B. Theoretical justification

Given that feature space corruption occurs under adversarial perturbations, we turn to attention for semantic identification. However, existing target-specific attention methods (Chefer et al., 2021; Gupta et al., 2022) remain vulnerable to adversarial attacks, primarily due to their inherent sensitivity to adversarial perturbations. We provide a theoretical proof using the PGD (Madry et al., 2018) attack algorithm as an example to demonstrate how adversarial perturbations exploit this sensitivity.

## B.1. Second-Order Sensitivity of Attention Gradients

**Lemma B.1.** *The variation of $\phi_u(\mathbf{x})$ with respect to the input $\mathbf{x}$ is governed by mixed second-order derivatives of $S(\mathbf{x})$.*

*Proof.* Fix an entry index $u$ of the attention tensor $\mathbf{A}^{(b)}(\mathbf{x})$, and define:

$$\phi_u(\mathbf{x}) \triangleq \left[\nabla_{\mathbf{A}^{(b)}(\mathbf{x})} S(\mathbf{x})\right]_u \triangleq \frac{\partial S(\mathbf{x})}{\partial (\mathbf{A}^{(b)}(\mathbf{x}))_u}. \tag{15}$$

Assuming local differentiability, for any small perturbation $\boldsymbol{\delta}$, the first-order Taylor expansion yields:

$$\phi_u(\mathbf{x} + \boldsymbol{\delta}) = \phi_u(\mathbf{x}) + \langle \nabla_{\mathbf{x}} \phi_u(\mathbf{x}), \boldsymbol{\delta} \rangle + O(\|\boldsymbol{\delta}\|_\infty^2), \tag{16}$$

where $\nabla_{\mathbf{x}}\phi_u(\mathbf{x})$ is a mixed second-order derivative of $S(\mathbf{x})$ with respect to the attention entry $(\mathbf{A}^{(b)}(\mathbf{x}))_u$ and the input $\mathbf{x}$:

$$\nabla_{\mathbf{x}}\phi_u(\mathbf{x}) = \frac{\partial^2 S(\mathbf{x})}{\partial(\mathbf{A}^{(b)}(\mathbf{x}))_u \partial \mathbf{A}^{(b)}(\mathbf{x})} \cdot \nabla_{\mathbf{x}}\mathbf{A}^{(b)}(\mathbf{x}). \tag{17}$$

Although the derivative in $\phi_u(\mathbf{x})$ is taken with respect to the attention entry $(\mathbf{A}^{(b)}(\mathbf{x}))_u$, its value is evaluated on the computation graph induced by the input $\mathbf{x}$. Hence, $\phi_u(\mathbf{x}) : \mathbb{R}^{d_x} \to \mathbb{R}$ is a well-defined function of $\mathbf{x}$, where $d_x$ denotes the dimensionality of the input. $\qquad\square$

**Corollary B.2.** *Under the constraint $\|\boldsymbol{\delta}\|_\infty \leq \varepsilon$, there exists a perturbation $\boldsymbol{\delta}^\star$ such that:*

$$|\phi_u(\mathbf{x} + \boldsymbol{\delta}^\star) - \phi_u(\mathbf{x})| = \varepsilon\|\nabla_{\mathbf{x}}\phi_u(\mathbf{x})\|_1 + O(\varepsilon^2). \tag{18}$$

*Therefore, if $\|\nabla_{\mathbf{x}}\phi_u(\mathbf{x})\|_1$ is non-negligible, $\phi_u(\mathbf{x})$ can undergo a $\Theta(\varepsilon)$ change. If, in addition, $(\mathbf{A}^{(b)}(\mathbf{x}))_u$ is bounded away from zero, then the corresponding entry of $\mathbf{A}^{(b)}(\mathbf{x}) \odot \nabla_{\mathbf{A}^{(b)}(\mathbf{x})}S(\mathbf{x})$ can also undergo a $\Theta(\varepsilon)$ change.*

*Proof.* By Hölder's inequality:
$$|\phi_u(\mathbf{x} + \boldsymbol{\delta}) - \phi_u(\mathbf{x})| \leq \varepsilon\|\nabla_{\mathbf{x}}\phi_u(\mathbf{x})\|_1 + O(\varepsilon^2), \tag{19}$$

the first-order upper bound is attained by:

$$\boldsymbol{\delta}^\star = \varepsilon \operatorname{sign}(\nabla_{\mathbf{x}}\phi_u(\mathbf{x})). \tag{20}$$

$\qquad\square$

Hence, the gradient component $\phi_u(\mathbf{x})$ can undergo an $O(\varepsilon)$ change under an $\ell_\infty$-bounded perturbation, driven by mixed second-order derivatives. Consequently, the corresponding entry of the gradient attention $\mathbf{A}^{(b)}(\mathbf{x}) \odot \nabla_{\mathbf{A}^{(b)}(\mathbf{x})}S(\mathbf{x})$ is also sensitive to $\ell_\infty$ perturbations.

## B.2. First-Order Stability of Token-Based Weights

**Lemma B.3.** *The rectified token-level gradient contribution $\left[\left\langle \mathbf{T}_v^{(b)}(\mathbf{x}), \mathbf{g}_v^{(b)}(\mathbf{x})\right\rangle\right]_+$ has local stability under bounded $\|\boldsymbol{\delta}\|_\infty \leq \varepsilon$ perturbations, and that this contribution implicitly aggregates local second-order information at the token level.*

*Proof.* The weight $\mathbf{W}^{(b)}(\mathbf{x})$ defined in Eq. (6) is derived from token-level gradient contributions. For the $v$-th token in the $b$-th layer, define:

$$\mathbf{g}_v^{(b)}(\mathbf{x}) \triangleq \nabla_{\mathbf{T}_v^{(b)}(\mathbf{x})}S(\mathbf{x}), \tag{21}$$

$$q_v^{(b)}(\mathbf{x}) \triangleq \left\langle \mathbf{T}_v^{(b)}(\mathbf{x}), \mathbf{g}_v^{(b)}(\mathbf{x})\right\rangle. \tag{22}$$

Let $\mathcal{U}_\varepsilon(\mathbf{x}) = \{\mathbf{x}' : \|\mathbf{x}' - \mathbf{x}\|_\infty \leq \varepsilon\}$ donates the perturbation neighborhood. Assume that $q_v^{(b)}$ is differentiable on $\mathcal{U}_\varepsilon(\mathbf{x})$ and that there exists a constant $L_v > 0$ satisfying:

$$\sup_{\mathbf{x}' \in \mathcal{U}_\varepsilon(\mathbf{x})} \left\|\nabla_{\mathbf{x}}q_v^{(b)}(\mathbf{x}')\right\|_1 \leq L_v. \tag{23}$$

Then, for any perturbation $\boldsymbol{\delta}$ satisfying $\|\boldsymbol{\delta}\|_\infty \leq \varepsilon$, define the rectified token-level gradient contribution as:

$$a_v^{(b)}(\mathbf{x}) = \left[q_v^{(b)}(\mathbf{x})\right]_+ \tag{24}$$

It must satisfies the following inequality:

$$\left|a_v^{(b)}(\mathbf{x} + \boldsymbol{\delta}) - a_v^{(b)}(\mathbf{x})\right| \leq L_v\varepsilon. \tag{25}$$

Since $q_v^{(b)}$ is differentiable on $\mathcal{U}_\varepsilon(\mathbf{x})$, the mean-value theorem gives the following expression for $\theta \in (0, 1)$:

$$q_v^{(b)}(\mathbf{x} + \boldsymbol{\delta}) - q_v^{(b)}(\mathbf{x}) = \left\langle \nabla_{\mathbf{x}}q_v^{(b)}(\mathbf{x} + \theta\boldsymbol{\delta}), \boldsymbol{\delta}\right\rangle. \tag{26}$$

By Hölder's inequality and $\|\boldsymbol{\delta}\|_\infty \leq \varepsilon$, we obtain:

$$\left| q_v^{(b)}(\mathbf{x} + \boldsymbol{\delta}) - q_v^{(b)}(\mathbf{x}) \right| \leq \left\| \nabla_{\mathbf{x}} q_v^{(b)}(\mathbf{x} + \theta\boldsymbol{\delta}) \right\|_1 \|\boldsymbol{\delta}\|_\infty \leq L_v \varepsilon. \tag{27}$$

Since the rectifier $[\cdot]_+$ is 1-Lipschitz, we have:

$$\left| a_v^{(b)}(\mathbf{x} + \boldsymbol{\delta}) - a_v^{(b)}(\mathbf{x}) \right| \leq \left| q_v^{(b)}(\mathbf{x} + \boldsymbol{\delta}) - q_v^{(b)}(\mathbf{x}) \right| \leq L_v \varepsilon. \tag{28}$$

Moreover, when $\mathbf{T}_v^{(b)}(\mathbf{x})$ and $\mathbf{g}_v^{(b)}(\mathbf{x})$ are differentiable with respect to $\mathbf{x}$, the gradient of $q_v^{(b)}$ satisfies:

$$\nabla_{\mathbf{x}} q_v^{(b)}(\mathbf{x}) = \left( \nabla_{\mathbf{x}} \mathbf{T}_v^{(b)}(\mathbf{x}) \right)^\top \mathbf{g}_v^{(b)}(\mathbf{x}) + \left( \nabla_{\mathbf{x}} \mathbf{g}_v^{(b)}(\mathbf{x}) \right)^\top \mathbf{T}_v^{(b)}(\mathbf{x}). \tag{29}$$

Then, the term $\nabla_{\mathbf{T}_v^{(b)}(\mathbf{x})} S(\mathbf{x})$ contains local second-order information of $S(\mathbf{x})$, but this information is aggregated into a token-level contribution rather than assigned to individual attention edges. $\square$

**Corollary B.4.** *Let the normalized token weight be defined as:*

$$W_v^{(b)}(\mathbf{x}) = \frac{a_v^{(b)}(\mathbf{x})}{\sum_{k=1}^s a_k^{(b)}(\mathbf{x})}. \tag{30}$$

*Assume that, for all $\mathbf{x}' \in \mathcal{U}_\varepsilon(\mathbf{x})$, the normalization denominator is bounded away from zero:*

$$\sum_{k=1}^s a_k^{(b)}(\mathbf{x}') \geq c_0 > 0. \tag{31}$$

*Also assume that each $a_k^{(b)}$ satisfies the bound in Lemma B.3. with constant $L_k$, and define:*

$$M_v = \sup_{\mathbf{x}' \in \mathcal{U}_\varepsilon(\mathbf{x})} a_v^{(b)}(\mathbf{x}'). \tag{32}$$

*Then, for any perturbation $\boldsymbol{\delta}$ satisfying $\|\boldsymbol{\delta}\|_\infty \leq \varepsilon$, the following inequality holds:*

$$\left| W_v^{(b)}(\mathbf{x} + \boldsymbol{\delta}) - W_v^{(b)}(\mathbf{x}) \right| \leq C_v \varepsilon, \tag{33}$$

*where $C_v$ is obtained as:*

$$C_v = \frac{L_v}{c_0} + \frac{M_v}{c_0^2} \sum_{k=1}^s L_k. \tag{34}$$

*Therefore, the normalized token weight $W_v^{(b)}(\mathbf{x})$ remains Lipschitz-stable within the $\ell_\infty$ perturbation neighborhood.*

*Proof.* Let define $Z(\mathbf{x})$ as:

$$Z(\mathbf{x}) = \sum_{k=1}^s a_k^{(b)}(\mathbf{x}). \tag{35}$$

Then the normalized token weight can be written as:

$$W_v^{(b)}(\mathbf{x}) = \frac{a_v^{(b)}(\mathbf{x})}{Z(\mathbf{x})}. \tag{36}$$

For $\|\boldsymbol{\delta}\|_\infty \leq \varepsilon$, we have:

$$|Z(\mathbf{x} + \boldsymbol{\delta}) - Z(\mathbf{x})| \leq \sum_{k=1}^s \left| a_k^{(b)}(\mathbf{x} + \boldsymbol{\delta}) - a_k^{(b)}(\mathbf{x}) \right| \leq \varepsilon \sum_{k=1}^s L_k. \tag{37}$$

Using $Z(\mathbf{x}) \geq c_0$ and $Z(\mathbf{x} + \boldsymbol{\delta}) \geq c_0$, we obtain:

$$
\begin{aligned}
\left| W_v^{(b)}(\mathbf{x} + \boldsymbol{\delta}) - W_v^{(b)}(\mathbf{x}) \right| &= \left| \frac{a_v^{(b)}(\mathbf{x} + \boldsymbol{\delta})}{Z(\mathbf{x} + \boldsymbol{\delta})} - \frac{a_v^{(b)}(\mathbf{x})}{Z(\mathbf{x})} \right| \\
&\leq \frac{\left| a_v^{(b)}(\mathbf{x} + \boldsymbol{\delta}) - a_v^{(b)}(\mathbf{x}) \right|}{Z(\mathbf{x} + \boldsymbol{\delta})} + a_v^{(b)}(\mathbf{x}) \left| \frac{1}{Z(\mathbf{x} + \boldsymbol{\delta})} - \frac{1}{Z(\mathbf{x})} \right| \\
&\leq \frac{L_v}{c_0}\varepsilon + \frac{M_v}{c_0^2} \left| Z(\mathbf{x} + \boldsymbol{\delta}) - Z(\mathbf{x}) \right| \\
&\leq \left( \frac{L_v}{c_0} + \frac{M_v}{c_0^2} \sum_{k=1}^{s} L_k \right) \varepsilon.
\end{aligned}
\tag{38}
$$

This proves the first-order stability of the normalized token-based gradient weight. $\square$

**Property 1.** *Under $\|\boldsymbol{\delta}\|_\infty \leq \varepsilon$ perturbations, the first-order variation of the column-scaled attention $\mathbf{B}^{(b)}(\mathbf{x})$ is additively composed of the attention variation and the token-weight variation, while token-level weighting avoids directly using the more sensitive edge-wise gradient attention term.*

*Proof.* Consider the column-scaling operation:

$$
\mathbf{B}^{(b)}(\mathbf{x}) = \mathbf{A}^{(b)}(\mathbf{x}) \operatorname{diag}\left( \mathbf{W}^{(b)}(\mathbf{x}) \right).
\tag{39}
$$

For a fixed attention head, its $(i, j)$-th entry is given by:

$$
B_{ij}^{(b)}(\mathbf{x}) = A_{ij}^{(b)}(\mathbf{x}) W_j^{(b)}(\mathbf{x}).
\tag{40}
$$

Let the perturbation operator be defined as:

$$
\Delta f = f(\mathbf{x} + \boldsymbol{\delta}) - f(\mathbf{x}).
\tag{41}
$$

Then the perturbation of $B_{ij}^{(b)}$ admits the exact decomposition:

$$
\Delta B_{ij}^{(b)} = \left( \Delta A_{ij}^{(b)} \right) \left( \Delta W_j^{(b)} \right) + \left( \Delta A_{ij}^{(b)} \right) W_j^{(b)}(\mathbf{x}) + A_{ij}^{(b)}(\mathbf{x}) \left( \Delta W_j^{(b)} \right).
\tag{42}
$$

If both $A_{ij}^{(b)}$ and $W_j^{(b)}$ vary at most linearly with $\varepsilon$ under $\|\boldsymbol{\delta}\|_\infty \leq \varepsilon$, then the perturbation can be written as:

$$
\Delta B_{ij}^{(b)} = \left( \Delta A_{ij}^{(b)} \right) W_j^{(b)}(\mathbf{x}) + A_{ij}^{(b)}(\mathbf{x}) \left( \Delta W_j^{(b)} \right) + O(\varepsilon^2).
\tag{43}
$$

Therefore, the first-order changes of $A_{ij}^{(b)}$ and $W_j^{(b)}$ enter additively. Unlike the edge-wise gradient attention term:

$$
A_{ij}^{(b)}(\mathbf{x}) \frac{\partial S(\mathbf{x})}{\partial A_{ij}^{(b)}(\mathbf{x})}.
\tag{44}
$$

$\square$

As a result, token-level column scaling is a more aggregated and more stable attention reweighting strategy.

## C. Experimental Supplements

### C.1. Setting Details

The main parameters and settings of A-TPT are provided in Table 8. Among them, the parameters of **Attention-guided Augmentation** are set as learnable on the validation set of ImageNet (Deng et al., 2009). After being determined through hyperparameter search, they are fixed and used for inference on the test set. The settings in **Prompt Tuning** follow previous

studies (Shu et al., 2022; Sheng et al., 2025), ensuring a unified and fair comparison. Notably, prompts are reset and tuned per samples after attacks, which is more consistent with real-world scenarios.

In our ResNet50 experiment, we use the final attention-pooling layer as the only available attention source. The pooled global query in this layer serves as the counterpart of the global token, and its attention over spatial locations is used as $\mathbf{A(x)}$. Moreover, the token-gradient weighting is retained only at the final attention-pooling layer, while the multi-layer rollout and last-2 averaging used in ViT are not applicable.

*Table 8.* Parameters and setting details of A-TPT

| Attention-guided Augmentation | | Prompt Tuning and Entropy Optimization | |
|---|---|---|---|
| Mixing Strength ($m_{high}$) | 0.8 | Learning Rate | 0.005 |
| Mixing Strength ($m_{low}$) | 0.2 | Learning Steps | 7 |
| Mask Ratio ($r$) | 0.2 | Number of Prompts | 4 |
| Number of Views $N$ | 64 | Ratio of Selected Viwes | 0.1 |

### C.2. Results Compared with Training-time Methods

In the main text of Sec 4.3, we compared A-TPT with training-time defense methods. Here, we provide a comprehensive evaluation of training-time methods on fine-grained datasets in Table 9 and Table 10 to highlight the efficiency of A-TPT, even in the absence of extra data. It is shown that A-TPT achieves the best performance both on clean and adversarial data.

*Table 9.* Clean accuracy across eight fine-grained datasets compared A-TPT with training-time defense methods (ViT-B/32).

| Method | Pets | Caltech101 | Cars | DTD | UCF101 | EuroSAT | Flower102 | Aircraft | Average |
|---|---|---|---|---|---|---|---|---|---|
| CLIP | 85.1 | 91.4 | 60.1 | 43.0 | 61.6 | 35.8 | 64.0 | 18.1 | 57.4 |
| TeCoA | 66.9 | 79.3 | 10.2 | 24.5 | 34.6 | 14.5 | 30.8 | 6.6 | 33.4 |
| APT+TeCoA | 66.7 | 81.4 | 20.8 | 35.2 | 40.2 | 29.3 | 42.5 | 5.2 | 40.2 |
| FARE | 76.7 | 86.3 | 39.2 | 28.3 | 44.2 | 16.6 | 37.0 | 9.5 | 42.2 |
| A-TPT | **87.3** | **92.0** | **63.8** | **44.0** | **63.4** | **35.9** | **64.5** | **17.5** | **58.6** |

*Table 10.* Adversarial accuracy across eight fine-grained datasets compared A-TPT with training-time defense methods (ViT-B/32).

| Method | Pets | Caltech101 | Cars | DTD | UCF101 | EuroSAT | Flower102 | Aircraft | Average |
|---|---|---|---|---|---|---|---|---|---|
| CLIP | 0.2 | 0.0 | 0.0 | 0.0 | 0.0 | 0.0 | 0.0 | 0.0 | 0.0 |
| TeCoA | 63.7 | 78.0 | 9.1 | 24.0 | 33.4 | 14.3 | 28.9 | 6.6 | 32.3 |
| APT+TeCoA | 63.9 | 80.2 | 18.9 | 33.7 | 39.4 | 29.2 | 40.4 | 5.2 | 38.9 |
| FARE | **73.8** | **85.4** | **34.4** | 27.3 | 41.9 | **16.3** | 34.0 | 9.5 | **40.3** |
| A-TPT | 66.4 | 79.8 | 31.8 | **31.1** | **46.9** | 12.7 | **43.2** | **10.4** | **40.3** |

