# A. Discussion

### A.1 Why attention-guided augmentation is necessary in test-time prompt tuning?

Existing test-time prompt tuning methods (Wang et al., 2025a; Sheng et al., 2025; Shu et al., 2022; Yoon et al., 2024) use multi-view augmentation to rewrite the optimization objective as an unsupervised expectation. They minimize entropy/uncertainty or enforce cross-view consistency, yielding more stable gradients that are less prone to bias from distribution shift or adversarial perturbations.

The goal of attention-guided augmentation is to generate multiple views that preserve discriminative regions under adversarial perturbations. More specifically:

- Random augmentations can easily introduce spurious background or texture noises, which will corrupt fine-grained discriminative parts.

- Attention, as a semantic anchor, can guide augmentation to preserve high-semantic regions while still introduce diversity. Preserving target parts or discriminative regions can effectively reduce the number of semantic-empty views.

- Attention-guided augmentation improves the stability of optimization by less low-quality views, and the effectiveness of ensemble by more high-quality views.

### A.2 Why is the semantic anchor attention instead of feature representations?

The key issue is not simply " Where semantic information are? ", but " Is there still unperturbed and discriminative semantics exist? ". Although semantics-preserving augmentation has been widely explored (Wang et al., 2021; Hu et al., 2024; Yang et al., 2025), existing problems in test-time adversarial satiation are:

- Adversarial attacks will push the features away from the decision boundary, which break the structure of the similarity or semantic neighborhood in feature space. In such case, using attacked feature representations to identify discriminative semantics is adding fuel to the fire.

- Attention is a training-independent signal. Directly learning semantics in feature space requires pre-learned prototypes, clustering, or dictionaries. As the regularization terms, it is difficult to decouple them from the training stage.

### A.3 Why does A-TPT need TV-based ensemble?

Not all views retain discriminative semantics under adversarial attacks. Since multiple views are generated from adversarially perturbed samples, some of them inherit aggressive adversarial noise, causing target parts to be broken or mixed with the background. Therefore, we need a way to filter out useless views. When a view has no clear discriminative regions, its attention map exhibits larger total variation. So the TV-based ensemble serves as the last line of defense.

# B. Theoretical justification

Given that feature space corruption occurs under adversarial perturbations, we turn to attention for semantic identification. However, existing target-specific attention methods (Chefer et al., 2021; Gupta et al., 2022) remain vulnerable to adversarial attacks, primarily due to their inherent sensitivity to adversarial perturbations. We provide a theoretical proof using the PGD (Madry et al., 2018) attack algorithm as a example to demonstrate how adversarial perturbations exploit this sensitivity.

**Lemma 1. Second-order sensitivity of attention gradients.** Fix any element index $p$ of the attention matrix $\mathbf{A}^{(b)}(x)$, and define the scalar function:

$$\phi_p(x) = \nabla_{\mathbf{A}^{(b)}(x)} S(x) = \frac{\partial S(x)}{\partial (\mathbf{A}^{(b)}(x))_p}. \tag{15}$$

Let $\phi_p(x)$ denote the gradient of the target logit $S(x)$ with respect to the $p$-th element of the attention matrix $\mathbf{A}^{(b)}(x)$. Its first-order derivative with respect to an intermediate variable $p$ is governed by mixed second-order derivatives of $S(x)$.

*proof.* Assuming local differentiability, for any small perturbation $\delta$, the first-order Taylor expansion yields:

$$\phi_p(x + \delta) = \phi_p(x) + \langle \nabla_x \phi_p(x), \delta \rangle + O(\|\delta\|^2), \tag{16}$$

Here, $\nabla_x \phi_p(x)$ corresponds to a mixed seconder-order derivative of $S(x)$ with respect to the attention variable and the input, as follow:

$$\nabla_x \phi_p(x) = \nabla_x \left( \frac{\partial S(x)}{\partial (\mathbf{A}^{(b)}(x))_p} \right) = \underbrace{\frac{\partial^2 S(x)}{\partial (\mathbf{A}^{(b)}(x))_p \partial \mathbf{A}^{(b)}(x)}}_{\text{Second-order (To Attention)}} \cdot \underbrace{\nabla_x \mathbf{A}^{(b)}(x)}_{\text{First-order (To Input } x)} . \tag{17}$$

Although the derivative in $\phi_p(x)$ is taken with respect to the attention element $(\mathbf{A}^{(b)}(x))_p$, the value of the derivative is evaluated on the computation graph induced by the adversarial $x$. Hence, $\phi_p(x) : \mathbb{R}^{d_x} \to \mathbb{R}$ is a well-defined function of $x$.

**Corollary 1. Instability of gradient attention under $\ell_\infty$ perturbations.** Under the constraint $\|\delta\|_\infty \leq \varepsilon$, there exists a perturbation $\delta^*$ such that the following equation holds:

$$|\phi_p(x + \delta^*) - \phi_p(x)| = \varepsilon \|\nabla_x \phi_p(x)\|_1 + O(\varepsilon^2). \tag{18}$$

Consider the gradient attention $\mathbf{A}^{(b)}(x) \odot \nabla_{\mathbf{A}^{(b)}} S(x)$, the corresponding entry can under go a $\Theta(\varepsilon)$ change.

*proof.* By Holder's inequality:

$$|\phi_p(x + \delta) - \phi_p(x)| \leq \varepsilon \|\nabla_x \phi_p(x)\|_1 + O(\varepsilon^2), \tag{19}$$

The upper bound of the inequality is attainable up to the first-order term by:

$$\delta^\star = \varepsilon \, \text{sign}(\nabla_x \phi_p(x)). \tag{20}$$

Hence, there exists an index $p$ s.t. $\|\nabla_x \phi_p(x)\|_1$ is non-negligible under an $\ell_\infty$-bounded perturbation, individual elements of this term can undergo $O(\varepsilon)$ changes driven by mixed second-order derivatives.

**Lemma 2. First-order sensitivity of token gradients.** The weight $\mathbf{W}^{(b)}(x)$ defined in Eq. (6) is derived from th token-level gradient. For the $q$-th token in the $b$-th layer, the weight takes the following form:

$$W_q^{(b)}(x) \propto \left\| \nabla_{T_q^{(b)}(x)} S(x) \right\|_2 . \tag{21}$$

Here, $\nabla_{T_q^{(b)}(x)} S(x)$ is a first-order derivative with respect to the token representation, and its variation with respect to the input exhibits the standard first-order Lipschitz behavior.

*proof.* Fix any token index $q$ of the embeddings in layer $b$, and define the scalar function:

$$\psi_q(x) \triangleq \nabla_{T_q^{(b)}} S(x), \tag{22}$$

Within a local differentiable neighborhood, for any small perturbation $\delta$, the first-order Taylor expansion yields:

$$\psi_q(x + \delta) = \psi_q(x) + \underbrace{\left( \nabla_x T_q^{(b)}(x) \right)^\top \nabla_{T_q^{(b)}}^2 S(x) \delta}_{\text{First-order Propagation Term}} + O\left( \|\delta\|^2 \right). \tag{23}$$

Here, the variation of $\psi_q(x)$ is from the first-order Jacobian $\nabla_x T_q^{(b)}(x)$ and the local Hessian $\nabla_{T_q^{(b)}}^2 S(x)$ with respect to the input embeddings. Hence, there is no mixed second-order derivative that couples different intermediate variables, and it just be a first-order propagation term.

**Corollary 2. Stability of token-gradient attention under $\ell_\infty$ perturbations.**

Let the scalar weight for the $q$-th token in the $b$-th layer be defined as:

$$W_q^{(b)}(x) = g \left( \nabla_{T_q^{(b)}(x)} S(x) \right), \quad g : \mathbb{R}^d \to \mathbb{R}. \tag{24}$$

Here, the aggregation function $g$ is Lipschitz continuous (*e.g.*, $\ell_1$, $\ell_2$, or other pooling). Under the same differentiability as in the Lemma 2, the following two properties hold.

**Property 1. Smoothing effect of vector-to-scalar aggregation** For a small perturbation $\delta$ with $\|\delta\| = O(\varepsilon)$:

$$|W_q^{(b)}(x + \delta) - W_q^{(b)}(x)| \leq C\varepsilon + O(\varepsilon^2), \tag{25}$$

Here, the constant $C$ depends only on the Lipschitz constant of $g$ and the operator norm of the token-input Jacobian $\nabla_x T_q^{(b)}(x)$. The aggregation over the embedding dimension suppresses the high-order coupling therms and acts as a low-pass filter, yielding a weight varies in a first-order Lipschitz manner with the input.

**Property 2. First-order stability of column-wise scaling.** Consider the column-scaling operation $\mathbf{A}^{(b)}(x)\operatorname{diag}\left(\mathbf{W}^{(b)}(x)\right)$, whose $q$-th entry $(i,j)$ is $\left(A^{(b)}(x)\right)_{ij} W_j^{(b)}(x)$. Its variation under the perturbation $\delta$ decomposes as:

$$\delta\left(A_{ij}^{(b)} W_j\right) = \left(\delta A_{ij}^{(b)}\right) W_j + A_{ij}^{(b)}\left(\delta W_j\right). \tag{26}$$

Here, the first-order change of $A_{ij}^{(b)}$ and the first-order change of the scalar token weight $W_j$ is independent. This additive decomposition shows that $\mathbf{A}^{(b)}\operatorname{diag}(\mathbf{W}^{(b)})$ eliminates the coupling between the attention matrix and the derivative of the loss with respect to the same matrix. Therefore, the column scaling operation does not introduce any new second-order amplification mechanism, and the overall propagation remains first-order stable.

## C. Experimental Supplements

### C.1. Setting Details

The main parameters and settings of A-TPT are provided in Table 6. Among them, the parameters of **Attention-guided Augmentation** are set as learnable on the validation set of ImageNet (Deng et al., 2009). After being determined through hyperparameter search, they are fixed and used for inference on the test set. The settings in **Prompt Tuning** follow previous studies (Shu et al., 2022; Sheng et al., 2025), ensuring a unified and fair comparison. Notably, prompts are reset and tuned per samples after attacks, which is more consistent with real-world scenarios.

*Table 6.* Parameters and setting details of A-TPT

| Attention-guided Augmentation | | Prompt Tuning and Entropy optimization | |
|---|---|---|---|
| Mixing Strength ($m_{high}$) | 0.8 | Learning Rate | 0.005 |
| Mixing Strength ($m_{low}$) | 0.2 | Learning Steps | 1 |
| Mask Ration ($r$) | 0.2 | Number of prompts | 4 |
| Number of Views $K$ | 64 | Ratio of Selected Viwes | 0.1 |

### C.2. Results Compared with Training-time Methods

In the main text of Sec 4.3, we compared A-TPT with training-time defense methods. Here, we provide a comprehensive evaluation of training-time methods on fine-grained datasets in Table 7 and Table 8 to highlight the efficiency of A-TPT, even in the absence of extra data. It is shown that A-TPT achieves the best performance both on clean and adversarial data.

*Table 7.* Clean accuracy (%) across 8 fine-grained datasets compared A-TPT with training-time defense methods (ViT-B/32).

| Method | Pets | Caltech101 | Cars | DTD | UCF101 | EuroSAT | Flower102 | Aircraft | Average |
|---|---|---|---|---|---|---|---|---|---|
| CLIP | 85.1 | 91.4 | 60.1 | 43.0 | 61.6 | 35.8 | 64.0 | 18.1 | 57.4 |
| TeCoA | 66.9 | 79.3 | 10.2 | 24.5 | 34.6 | 14.5 | 30.8 | 6.6 | 33.4 |
| APT+TeCoA | 66.7 | 81.4 | 20.8 | 35.2 | 40.2 | 29.3 | 42.5 | 5.2 | 40.2 |
| FARE | 76.7 | 86.3 | 39.2 | 28.3 | 44.2 | 16.6 | 37.0 | 9.5 | 42.2 |
| A-TPT | **87.3** | **92.0** | **63.8** | **44.0** | **63.4** | **35.9** | **64.5** | 17.5 | **58.6** |

*Table 8.* Adversarial accuracy (%) across 8 fine-grained datasets compared A-TPT with training-time defense methods (ViT-B/32).

| Method | Pets | Caltech101 | Cars | DTD | UCF101 | EuroSAT | Flower102 | Aircraft | Average |
|---|---|---|---|---|---|---|---|---|---|
| CLIP | 0.2 | 0.0 | 0.0 | 0.0 | 0.0 | 0.0 | 0.0 | 0.0 | 0.0 |
| TeCoA | 63.7 | 78.0 | 9.1 | 24.0 | 33.4 | 14.3 | 28.9 | 6.6 | 32.3 |
| APT+TeCoA | 63.9 | 80.2 | 18.9 | 33.7 | 39.4 | 29.2 | 40.4 | 5.2 | 38.9 |
| FARE | **73.8** | **85.4** | **34.4** | 27.3 | 41.9 | **16.3** | 34.0 | 9.5 | **40.3** |
| A-TPT | 66.4 | 79.8 | 31.8 | **31.1** | **46.9** | 12.7 | **43.2** | **10.4** | **40.3** |