# OpenReview forum: "Towards Fine-Grained Robustness: Attention-Guided Test-Time Prompt Tuning for Vision-Language Models"
_ICML.cc/2026/Conference — ICML 2026 regular_

### Official Review · Reviewer_dZHg · 2026-02-28

**Soundness:** 3
**Presentation:** 3
**Significance:** 3
**Originality:** 3
**Overall Recommendation:** 4
**Confidence:** 3

**Summary:**

The paper observes that previous test-time prompt tuning frameworks typically rely on optimization within the feature space. However, the authors identify that the feature space exhibits unreliability under adversarial attacks whereas the attention map remains reliable. Consequently, they propose performing a series of operations on the attention map to enhance fine-grained robustness. The proposed method achieves state-of-the-art performance on ImageNet and eight fine-grained benchmark datasets while ensuring that performance on clean data is preserved.

**Compliance With Llm Reviewing Policy:**

Affirmed.

**Final Justification:**

I maintain my weak accept. The paper proposes a reasonable shift from feature-space to attention-map-based optimization for adversarial robustness. The rebuttal adequately addressed my concerns on inference efficiency and attention map quality. Overall, the contribution is solid but incremental, which aligns with my original assessment.

**Key Questions For Authors:**

First, since the proposed method falls under the category of Test-Time Adaptation where inference latency constitutes a core bottleneck, the manuscript would be more persuasive if the authors provided a comparison of inference times across different methods.

Second, given that the method utilizes attention maps to guide the data augmentation process, the accuracy of these attention maps is critical, so it would be beneficial to conduct a quantitative experiment to evaluate the extent to which the attention maps cover the target regions.

A minor weakness is observed regarding inconsistencies in information across the paper, such as the discrepancy between DTD and DAD in Figure 1.

**Limitations:**

Yes

**Strengths And Weaknesses:**

- **Soundness**: The paper first analytically demonstrates the unreliability of the feature space employed by prior methods and subsequently shifts focus toward seeking reliable attention maps. A rigorous set of mathematical derivations is provided in the supplementary material to support this claim. The experimental design is comprehensive, covering eight fine-grained datasets along with ImageNet, and the comparison baselines include the latest state-of-the-art methods from 2025.

- **Clarity**: The manuscript is well-organized and clearly structured.

- **Significance**: This work presents a novel solution to an existing problem.

- **Originality**: The originality of this work primarily lies in its shift of perspective and cross-domain integration. Its key novelty stems from moving away from the feature-space-centric view emphasized by previous approaches toward an alternative perspective, thereby broadening the scope of potential problem-solving strategies.

---

> ### Author Rebuttal · Authors · 2026-03-30
>
> **Q1: Inference Times**
>
> Thanks for the suggestion.
>
> We further study the inference efficiency, finding that the additional computational cost stems from per-sample attention extraction, as shown in Table 4. Moreover, the inference can be accelerated by reducing the number of views to obtain better real-time performance. By reducing this number from 64 to 16, the time required for per-sample is reduced to less than 1/5 of the original, but the accuracy remains stable.
>
> Compared with previous studies such as RTPT whose adversarial accuracy is 60.2% with running time of 0.58s/per-sample, A-TPT exhibits strong accuracy–efficiency trade-off.
>
> Table 1. Inference efficiency on the Pets dataset (ViT-B/16).
>
> | Views Number | Time/per-sample | Accuracy |
> |:-:|:-:|:-:|
> |64|2.71 s|70.5|
> |32|1.07 s|70.1|
> |16|0.53 s|69.8|
>
> **Q2: Quantitative Experiments of Attention**
>
> Thanks for the suggestion.
>
> We visualize the quality of several attention maps in Figure 3 of the manuscript, where it can be observed that different views consistently share the same fine-grained object part. We further define Mass@Obj to quantify the proportion of the total attention mass that falls within the target object region:
> $$
> \\mathrm{Mass@Obj}(A, G) = \\frac{\\sum_{u,v} A_{u,v} G_{u,v}}{\\sum_{u,v} A_{u,v}},
> $$
> where $A$ is an attention map and $G$ is an object region mask.
>
> Additional experiments on Pets and Aircraft datasets are reported in Table 2. The results show that the refined attention can accurately localize the target region. Notably, although Aircraft achieves higher localization accuracy, its classification performance is still much lower than that on Pets. We believe this is because the fine-grained parts that contribute to discrimination within the aircraft meta-class (e.g., wings and tails) are much more similar across categories, whereas the shared fine-grained parts within the pet meta-class (e.g., eyes and fur) exhibit larger inter-category differences.
>
> Table 2. Adversarial Mass@Obj under PGD attacks (ViT-B/32).
>
> |               Mass@Obj             | Pets | Aircraft |
> |:-------------------------:|:----:|:--------:|
> | Average       | 0.73 | 0.82     |
> | 95th percentile  | 0.88 | 0.91     |
> | 5th percentile| 0.48 | 0.51     |
>
> **Q3: Spelling Error**
>
> Thanks for your correction, and it will be corrected in the final version.

---

> > ### Author Rebuttal · Reviewer_dZHg · 2026-04-01
> >
> > My concerns have been adequately addressed.

---

> > > ### Author Response · Authors · 2026-04-01
> > >
> > > Thanks for your reply.

---

### Official Review · Reviewer_AGEx · 2026-03-07

**Soundness:** 3
**Presentation:** 3
**Significance:** 3
**Originality:** 3
**Overall Recommendation:** 4
**Confidence:** 2

**Summary:**

This paper proposes a token-based gradient signal refinement method against adversarial attacks, called A-TPT, which consists of two core modules: Attention-guided multi-view Augmentation and TV-based Ensemble. A wide range of experiments on fine-grained tasks show that A-TPT not only identifies and preserves discriminative regions but also outperforms SOTA compared with other test-time adaptation methods.

**Compliance With Llm Reviewing Policy:**

Affirmed.

**Key Questions For Authors:**

- **(Q1)** Could the authors explain the “?” meanings?
- **(Q2)** Could the authors supply more information about the augmentation methods in the paper?
- **(Q3)** Could the authors add some results from large-scale datasets for this method?
- **(Q4)** Could the author further explore the detailed augmentation methods in this paper, for example, some color enhancement methods, or some mixup-based methods?

**Limitations:**

Yes

**Strengths And Weaknesses:**

### Strengths
- This paper proposes a novel method for countering adversarial attacks and improving zero-shot ability on classification tasks.

- A wide range of experiments from 9 fine-grained datasets, results demonstrate that A-TPT outperforms comparable methods.


### Weaknesses
- **(W1)** Figures 2 & 4 show some “?”, so the authors should check why it happened. If not, please explain what it means.

- **(W2)** Some detailed information was not explained in the paper: which augmentation method was used in this method.

- **(W3)** This paper does not evaluate on some large-scale datasets, like ImageNet-1K. And what are the prompt templates?

- **(W4)** The ablation study is insufficient. It’s not clear which augmentation method was used and whether it was truly useful for this method.

---

> ### Author Rebuttal · Authors · 2026-03-30
>
> **W1 & Q1: Symbol "?"**
>
> We checked the PDF displayed in the system and confirmed the presence of the “ ? ” symbols you mentioned. This seems to be caused by a rendering issue of the vector graphics in the system. Specifically, in Figure 2, the “ ? ” symbols should be $p(x_0)$, $p(x_1)$, $p(x_N)$, and $TV(A(x_i)) \\rightarrow \\omega_i$ , while in Figure 4, they should be $\\omega > 0.1$ and $\\omega < 0.01$.
>
> **W2 & Q2: Augmentation Method**
>
> Thanks for the question. We state on line 230 of page 5 in the manuscript that the augmentation is based on AugMix [1]. However, the corresponding citation was omitted, and we will correct this in the final version.
>
> **W3 & Q3: Results on Large-scale Datasets**
>
> Thanks for the suggestion. We evaluate A-TPT on one ImageNet dataset as shown in Table 1 and Table 2 in the manuscript, and more results on large-scale datasets are meaningful.
>
> Following the experimental setting used by previous studies [2,3] on the ImageNet-OOD datasets, we use ResNet50 as the backbone and evaluate robustness under PGD with $\epsilon$ = 1/255 and 7 steps. As shown in Table 1, A-TPT is compared with the SOTA method RTPT, and still achieves stronger robustness on the ImageNet-OOD datasets.
>
> Due to time constraints, we only report experimental results on ImageNet-A and ImageNet-V2, and the complete results will be included in the final version.
>
> Table 1. Adversarial accuracy on ImageNet-A and ImageNet-V2 (ResNet50).
>
> |Methods|ImageNet-A|ImageNet-V2|
> |:-:|:-:|:-:|
> |RTPT|14.4|41.6|
> |A-TPT|16.1|44.2|
>
> **W4 & Q4: Detailed Augmentation Methods**
>
> Thanks for the comment.
>
> AugMix [1] first applies preprocessing to obtain the base view $\mathrm{base}$, where the preprocessing consists of *Random Resized Crop* and *Random Horizontal Flip*. Then three augmentation chains are sampled independently. For the $i$-th chain, it starts from the base view and randomly chooses a chain length between 1 and 3 steps. At each step, one augmentation operator is randomly selected from: *Auto Contrast*, *Histogram Equalization*, *Posterization*, *Rotation*, *Solarization*, *Horizontal Shear*, *Vertical Shear*, *Horizontal Translation*, *Vertical Translation*.
>
> After the three chains are applied, it obtains three augmented images and first computes a convex combination:
>
> $$
> x = w_1 \\cdot \\mathrm{chain}_1 + w_2 \\cdot \\mathrm{chain}_2 + w_3 \\cdot \\mathrm{chain}_3
> $$
>
> where $w = (w_1, w_2, w_3)$ is sampled from $ \\mathrm{Dirichlet}(1,1,1) $, so all weights are non-negative and sum to 1. Then a global mixing coefficient $m \\sim \\mathrm{Beta}(1,1)$ is sampled, and the base image is interpolated with the mixed augmentation as:
>
> $$
> \\hat{x} = m \\cdot \\mathrm{base} + (1-m) \\cdot x
> $$
>
> In our method, we replace the scalar $m$ with spatially varying coefficients determined by attention, namely $m_{high}$ and $m_{low}$.
>
> **References**
>
> [1] Hendrycks, D., Mu, N., Cubuk, E. D., Zoph, B., Gilmer, J., and Lakshminarayanan, B. AugMix: A Simple Data Processing Method to Improve Robustness and Uncertainty. In Proc. Int. Conf. Learn. Representations, pp. 1–15, 2020.
>
> [2] Sheng, L., Liang, J., Wang, Z., and He, R. R-TPT: Improving Adversarial Robustness of Vision-Language Models through Test-Time Prompt Tuning. In Proc. IEEE Conf. Comp. Vis. Patt. Recogn., pp. 29958–29967, 2025.
>
> [3] Zanella, M. and Ben Ayed, I. On the test-time zero-shot generalization of vision-language models: Do we really need prompt learning? In Proc. IEEE Conf. Comp. Vis. Patt. Recogn., pp. 23783–23793, 2024.

---

> > ### Author Rebuttal · Reviewer_AGEx · 2026-04-01
> >
> > My concerns have been addressed.

---

> > > ### Author Response · Authors · 2026-04-01
> > >
> > > Thanks for your reply.

---

### Official Review · Reviewer_q6Sb · 2026-03-10

**Soundness:** 2
**Presentation:** 2
**Significance:** 2
**Originality:** 2
**Overall Recommendation:** 4
**Confidence:** 4

**Summary:**

This paper proposes Attention-guided Test-time Prompt Tuning (A-TPT), a test-time defense method for CLIP-based vision-language models under adversarial attacks, with a focus on fine-grained recognition. The core idea is to replace the gradient signal in Gradient Attention Rollout (GAR) with a token-gradient-based weighting that is theoretically more stable under perturbations, and then use the resulting attention maps to guide spatially varying augmentation that preserves discriminative regions, and weight views via anisotropic Total Variation for final ensemble. Experiments on ImageNet and eight fine-grained datasets show that A-TPT achieves the best average adversarial accuracy among the compared test-time methods while also improving clean accuracy under a PGD attack protocol.

**Compliance With Llm Reviewing Policy:**

Affirmed.

**Final Justification:**

I thank the authors for the thorough follow-up. The Mass@Obj analysis, TV reduction across five datasets, and cross-condition attention transfer experiment address my main concerns. I raise my score from Weak Reject to Weak Accept and recommend incorporating these analyses into the final version.

**Key Questions For Authors:**

Please refer to Weaknesses above.

**Limitations:**

The paper lacks a limitations section. For example, the computational overhead of per-sample attention extraction, 64-view augmentation, and TV scoring at test time, is not discussed.

**Strengths And Weaknesses:**

* Strengths
	* The problem is well-motivated. The paper identifies a limitation of existing test-time methods that random augmentation can destroy discriminative regions in fine-grained tasks and the proposed direction of using attention as a semantic anchor is a reasonable approach.
	* The three components (Attention Refinement, Attention-guided Augmentation, TV-based Ensemble) form a coherent pipeline, and the ablation in Table 5 shows that the components are interdependent. A-Refine along gives no gain, but enables A-Aug and A-TV to function, which is informative about how the proposed method works.
	* Within the reported PGD protocol, A-TPT shows consistent improvement over the compared test-time baselines on both adversarial and clean accuracy across two backbones.
* Weaknesses
	* The most critical component, Gradient Attention Refinement, contains multiple under-justified design choices. Since the entire pipeline (augmentation, view selection, ensemble) depends on the quality of this attention map, these gaps undermine the foundation of the method
		* The last-2 layer averaging in Eq. (8) is motivated only by a one-line claim about suppressing shallow-layer noise, with no ablation over the number of layers.
		* Eq. (8) averages a refine transition matrix (hatted) with a raw attention matrix (unhatted), and the final rollout $\hat{A}_{(x)}$ double-counts the last layer, but whether this asymmetry is intentional is unexplained.
		* The notation is ambiguous enough that it is unclear whether the final rollout is matrix multiplication or element-wise multiplication.
	* The attention refinement in Eq. (6) relies on the gradient of a target logit $S(x)$, but the paper does not specify how this target is chosen. Under adversarial attack, the top-1 prediction is likely wrong (as Figure 1(b) shows), so the attention map may highlight regions discriminative for the wrong class. This issue correlates with the data: Aircraft has only 0.8% true label in top-10 and the lowest accuracy (15.1%), while Pets has 31.4% and much higher accuracy (70.5%).
	* The main benchmark uses only PGD. CW, DeepFool, and FGSM are evaluated on just two datasets (Flower, DTD) with one backbone (ViT-B/32). Notably, TAPT (Wang et al., 2025a), a directly comparable test-time adversarial defense method that is discussed in the related work but omitted from the comparison, employs stronger attacks such as AutoAttack. The absence of both TAPT as a baseline and stronger attack protocols weakens the robustness claims.
	* In Eq. (11), base view already contain the adversarial perturbation, and the spatially varying mixing retains much of the base view in high-attention areas. The paper frames this as semantic protection, but it simultaneously preserves adversarial noise in the regions most critical for classification. This trade-off is not analyzed.
	* Minor issues
		* In page 8, A-TV clean accuracy gain is stated as +4.8% but Table 5 shows +4.0%, and A-Aug gain is stated as +5.3% but Table 5 shows +4.3%. Also, final accuracy is stated as 62.8% but Table 5 shows 63.0%.
		* Multiple typos: line 150 (We -> we), line 156 (Sec. 2 -> Sec. 3.2), line 252 (employ -> employed), line 648 (a example -> an example), Figure 2 caption (input simple -> input sample)
		* Were the augmentation hyperparameters (m_high, m_low, r) tuned via grid search on ImageNet validation accuracy? How sensitive is performance to these values across different fine-grained datasets?

---

> ### Author Rebuttal · Authors · 2026-03-30
>
> Thanks for your comments. Due to time and character constraints, we report additional experiments (PGD with $\\epsilon$ = 4/255 and 100 steps) only on the Pets and Aircraft datasets here, and the complete results will be included in the final version.
>
> **W1.1: Ablation on the Number of Layers**
>
> The last-2 layer averaging is motivated by previous studies [1,2]. They show that semantic part information emerges most in deep ViT layers. Although filtering can reduce shallow-layer noise, the additional computational overhead makes it less suitable for efficient test-time adaptation. We conducted an ablation study on attention layers, as shown in Table 1.
>
> Table 1. Adversarial accuracy with different numbers of attention layers (ViT-B/32).
>
> |Layers Number|Pets|Aircraft|
> |-|:-:|:-:|
> | All Layers|57.2|5.3|
> | Last-three Layers|60.6|8.7|
> | Last-two Layers|62.7|9.1|
> | Last-two-avg Layers|66.4|10.4|
>
> **W1.2: Design of Eq. (8)**
>
> $A^{(B)}$ is indeed an error in Eq. (8), and the correct should be $\\hat{A}^{(B)}$. The final rollout in Eq. (8) is expanded as:
> $$
> \\hat{A} = \\frac{1}{2}\\hat{A}^{(B)}\\hat{A}^{(B-1)} + \\frac{1}{2}(\\hat{A}^{(B)})^2
> $$
> The final attention is used not only for semantic localization, but for generating a semantic mask. The first term $\\frac{1}{2}\\hat{A}^{(B)}\\hat{A}^{(B-1)}$ is introduced to identify continuous and smooth semantic regions, and only regions that form a valid pathway across both layers $B-1$ and $B$ are retained. The second term $\\frac{1}{2}(\\hat{A}^{(B)})^2$ drives the attention distribution to contract toward the most confident regions, because regions with strong responses in the last layer are further strengthened. (as shown in Table 1)
>
> **W1.3: Matrix Multiplication**
>
> The final rollout uses matrix multiplication, since it aims to capture the accumulated token-to-token propagation across layers as described in Eqs. (21)–(23).
>
> **W2: $S(x)$ under Shared Fine-grained Structures**
>
> We indeed didn’t clearly state that the target logit $S(x)$ is computed from the top-1 prediction. However, we would like to clarify that this doesn’t imply that the final attention highlights regions for the wrong class.
>
> In our method, attention is not used for the final class prediction, but only for fine-grained part localization. Classical FGVC studies [1,2] show that within the same meta-class, different fine-grained categories often share a relatively stable object-part layout and structural scaffold, and the features tend to cluster by parts rather than by class on these shared structures. Therefore, the top-1 prediction is used only to reliably identify the meta-class, and the resulting attention map can remain object-centric and provide a useful semantic anchor for part localization.
>
> Additional quantitative analysis please refer to our **”Q2: Quantitative Experiments of Attention”** response to Reviewer 4.
>
> **W3: Comparisons with TAPT under AutoAttack**
>
> We added comparisons with TAPT and further evaluate model robustness under AutoAttack. As shown in Table 2, under the same attack settings reported in TAPT, A-TPT performs strong robustness.
>
> Table 2. Adversarial accuracy compared with TAPT (ViT-B/32)
>
> |Methods|Pets (PGD)|Aircraft (PGD)|Pets (Auto)|Aircraft (Auto)|
> |:-:|:-:|:-:|:-:|:-:|
> |TAPT|68.1|4.5|68.4|5.0|
> |A-TPT|70.6|13.2|71.2|13.8|
>
> **W4: Sensitivity Analysis of Trade-off**
>
> We performed a sensitivity analysis of mixing strengths $\\lambda(r) \\in \\{m_{high},\\ m_{low}\\}$, as shown in Table 3. The results show that A-TPT is not sensitive to the parameters controlling this trade-off, indicating that although the discriminative regions selected by $S(x)$ still contain a small amount of adversarial noise, they indeed preserve most of the intact semantic information.
>
> Table 3. Adversarial accuracy with different trade-off mixing strengths (ViT-B/32).
>
> | Mixing Strengths | Pets | Aircraft |
> |:-:|:-:|:-:|
> |9:1|65.8|10.1|
> |8:2|66.4|10.4|
> |7:3|64.2|10.1|
>
> **W5: Minor Issues**
>
> Thanks for your corrections, and we will correct them in the final version. Mixing strengths are analyzed in **W4: Sensitivity Analysis of Trade-off**.
>
> **Limitations**
>
> Thanks for your comment. Due to character constraints, please refer to the discussion of *Inference Efficiency* in our **”W5: Inference Efficiency”** response to Reviewer 1 and the discussion of *Limitations* in our **”W4 & Limitations”** response to Reviewer 1.
>
> **References**
>
> [1]Caron, M., Touvron, H., Misra, I., Jégou, H., Mairal, J., Bojanowski, P., and Joulin, A. Emerging Properties in Self-Supervised Vision Transformers. In Proc. IEEE/CVF Int. Conf. Comp. Vis., pp. 9650–9660, 2021.
>
> [2] Krause, J., Gebru, T., Deng, J., Li, L.-J., and Fei-Fei, L. Learning Features and Parts for Fine-Grained Recognition. In Proc. Int. Conf. Pattern Recogn., pp. 26–33, 2014.

---

> > ### Author Rebuttal · Reviewer_q6Sb · 2026-04-01
> >
> > Thank you for the detailed rebuttal and the additional experiments. Below are my remaining concerns.
> >
> > * W1 response: Reference [1] studies self-supervised ViT properties and [2] addresses CNN-based part learning for fine-grained recognition. Neither prescribes the last-2-averaging design in Eq. (8), and [2] is not even based on ViT architectures. These references do not support the proposed design choice. Moreover, the ablation shows high sensitivity to layer selection, which warrants a deeper analysis of the components of the proposed method.
> > * W2 response: This argument assumes that adversarial misclassifications stay within the same meta-class, but no evidence is provided. Furthermore, my original review pointed out a clear correlation between top-K true label ratio and final accuracy (for Pets and Aircraft datasets), which the response does not address. This dataset-level pattern suggests that when the attack pushes predictions far from the true class, the attention anchor becomes unreliable, which is more consistent with my concern than the shared-structure explanation.
> > * The rebuttal experiments are all limited to Pets and Aircraft. This is insufficient to support general conclusions.

---

> > > ### Author Response · Authors · 2026-04-03
> > >
> > > **1. Response to W1**
> > >
> > > **1.1  Ablation Study and Mechanistic Analysis**
> > >
> > > Thanks for the comments. We agree that [1] and [2] do not by themselves prescribe the exact last-two-averaging form in Eq. (8). We would like to clarify that they are not cited as a direct derivation of our rollout operator, but only to motivate two general principles:
> > >
> > > (1) semantic information is concentrated in deeper representations.
> > >
> > > (2) fine-grained recognition benefits from preserving discriminative parts.
> > >
> > > The direct evidence for Eq. (8) comes from the layer ablation study: using all layers performs worst, restricting to the last three and last two layers improves adversarial accuracy, and the proposed last-two averaging performs best. Importantly, this sensitivity is not arbitrary, but an explainable and systematic layer effect.
> > >
> > > Moreover, what makes last-two-averaging design perform better is:
> > >
> > > (1) The first term $\\frac{1}{2}\\hat{A}^{(B)}\\hat{A}^{(B-1)}$ of extended Eq. (8) encourages cross-layer consistent semantic paths, retaining regions that form a valid pathway across both layers $B-1$ and $B$.
> > >
> > > (2) The second term $\\frac{1}{2}(\\hat{A}^{(B)})^2$ strengthens high-semantic regions in the last layer, driving the attention distribution to contract toward the most confident regions.
> > >
> > > **1.2 Further Analysis**
> > >
> > > Firstly, we define Mass@Obj to quantify the proportion of the total attention mass that falls within the target object region:
> > > $$
> > > \\mathrm{Mass@Obj}(A, G) = \\frac{\\sum_{u,v} A_{u,v} G_{u,v}}{\\sum_{u,v} A_{u,v}},
> > > $$
> > > where $A$ is an attention map and $G$ is an object region mask.
> > >
> > > As reported in Table 1, shallow-layer attention contributes substantial off-object noise, while deeper layers are markedly cleaner and more object-centric. Therefore, the proposed last-two-avg design is not only an empirical choice, but the practical best-performing variant among the tested layer combinations.
> > >
> > > Table 1. Mass@Obj with different numbers of attention layers under PGD attacks (ViT-B/32).
> > >
> > > |Layers|Pets|Aircraft|
> > > |:-:|:-:|:-:|
> > > |All Layers| 53.10%|57.37%|
> > > |Last-three Layers|68.22%|79.22%|
> > > |Last-two Layers|72.16%|82.16%|
> > > |Last-two-avg Layers|73.37%|82.32%|
> > >
> > > Secondly, why the last-two-avg layers perform better than the last-two layers is also important. We use anisotropic Total Variation (TV), described in Eq. (12) of the manuscript, to quantify the spatial fragmentation of an attention map. We then compare the relative TV reduction from the last-two layers to the last-two-avg layers, defined as:
> > >
> > > $$
> > > \\Delta TV = \\frac{TV_{\\text{Last-two}} - TV_{\\text{Last-two-avg}}}{TV_{\\text{Last-two}}}
> > > $$
> > >
> > > Although their Mass@Obj are similar, the last-two-avg design consistently produces a less fragmented attention map than the Last-two design, as shown in Table 2. Therefore, beyond achieving slightly higher Mass@Obj, the last-two-avg layers are also preferable because they yield less fragmented responses.
> > >
> > > Table 2. Relative TV reduction from the last-two layers to the last-two-avg layers under PGD attacks (ViT-B/32).
> > >
> > > |Pets|Aircraft|Cars|Caltech101|Flower102|
> > > |:-:|:-:|:-:|:-:|:-:|
> > > |9.21%|6.60%|7.64%|10.3%|9.13%|
> > >
> > > These suggest that the design is not only empirical, but aligned with the semantic concentration of deep layers and the need for cross-layer consistency.
> > >
> > > **2. Response to W2**
> > >
> > > Thanks for the comments. The statement that different fine-grained categories within the same meta-class often share a relatively stable object-part layout is based on specific studies on this problem. This motivates our view that the role of $S(x)$ is only to localize semantic anchors for mask construction, not to determine the final class label.
> > >
> > > To directly assess whether the adversarial semantic anchor remains reliable, we transfer the attention extracted from adversarial images to mask their corresponding clean images, and compare the resulting accuracy, as shown in Table 3. The results show that, although adversarial attacks affect the localization of semantic anchors slightly, they remain reliable.
> > >
> > > Table 3. Clean accuracy (%): comparing adversarial anchors localization with clean anchors localization (ViT-B/32).
> > > |Methods|Pets|Aircraft|Cars|Caltech101|Flower102|
> > > |:-:|:-:|:-:|:-:|:-:|:-:|
> > > |$Anchor_{clean}$ for $Image_{clean}$|87.3|17.5|63.8|92.0|64.5|
> > > |$Anchor_{adv}$ for $Image_{clean}$|86.7|17.1|63.2|91.4|63.6|
> > > |No Anchor|75.0|16.4|51.7|88.2|58.1|
> > >
> > > Regarding the correlation between the top-K true-label ratio and final accuracy, we do not think it should be interpreted as direct evidence that the semantic anchors are incorrectly localized. As shown in the third row of Table 3, it is more consistent with the fact that the similarity levels of the shared fine-grained parts within each meta-class vary substantially across datasets.
> > >
> > > Additionally, Figure 1 of the manuscript only reports the top-K true-label ratio for a single attacked sample, whereas our final prediction is based on selecting the top 10% most reliable views among augmented views.

---

### Official Review · Reviewer_37JE · 2026-03-13

**Soundness:** 3
**Presentation:** 2
**Significance:** 2
**Originality:** 3
**Overall Recommendation:** 4
**Confidence:** 4

**Summary:**

This paper proposes A-TPT, a method for test-time adversarial adaptation of VLMs.  The method first refines Gradient Attention Rollout (GAR) by replacing the original attention-based gradient signal with a token-based gradient signal. Based on the refined attention maps, Attention-guided Multi-view Augmentation generates augmented views with spatially varying augmentation strengths to better preserve discriminative regions. And TV-based Ensemble is proposed to measure the spatial fragmentation of attention maps and assign weights to different views during prediction aggregation.

**Compliance With Llm Reviewing Policy:**

Affirmed.

**Final Justification:**

I appreciate the authors’ efforts during the rebuttal and would like to maintain my original score. I hope the authors can include the complete experimental results in the revised manuscript.

**Key Questions For Authors:**

- Is the symbol $t$ in $\mathrm{logits}_t(x)$ the same as the $t$ used in Formula (3)?
- Why can the proposed Attention Refinement accurately identify unperturbed fine-grained parts in nature, given that adversarial perturbations are applied to essentially all pixels?
- Why is the result on Aircraft different from the one reported in R-TPT, while most of the other reported results appear to be the same?
- Are the results in Tables 1 and 2 based on zero-shot CLIP with test-time adaptation? I suggest the authors clarify this explicitly in the tables or captions.

**Limitations:**

please provide the limitation

**Strengths And Weaknesses:**

### strength

- The paper is generally well organized and easy to follow.
- The results appear impressive.
- The code is provided, which improves reproducibility

### weakness

- The method introduces several additional hyperparameters, but the paper does not provide sufficient sensitivity analysis.
- Experiments are restricted to CLIP ViT-B backbones, so it remains unclear whether the proposed method generalizes to other architectures, such as CLIP-ResNet50.
- Generalization experiments are limited and less comprehensive than prior work such as R-TPT and AOM. In particular, evaluation on ImageNet-OOD benchmarks (ImageNet-A/ImageNet-V2/ImageNet-R/ImageNet-S) is missing. It would also be valuable to add a comparison under AutoAttack, as conducted in AOM.
- The paper lacks adaptive attack analysis specifically targeting the proposed attention refinement and Attention-guided Multi-view Augmentation. More broadly, robustness is mainly validated under image-side attacks. It would be valuable to demonstrate effectiveness against multimodal attacks, such as Co-Attack and VLATTACK.
- Inference cost analysis is not provided.

---

> ### Author Rebuttal · Authors · 2026-03-30
>
> Thanks for all your comments. Due to time and character constraints, additional analysis and experiments are reported only on representative datasets in this rebuttal, and the complete results will be included in the final version.
>
> **W1: More Sensitivity Analysis**
>
> A-TPT does introduce additional parameters, and the number of views $N$ is analyzed in Figure 5 of the manuscript. In addition, the mixing strengths $\\lambda(r) \\in \\{m_{high},\\, m_{low}\\}$ are studied in Table 1. The results show that A-TPT remains robust whether the high-attention regions are relatively over-preserved (first row) or insufficiently preserved (second row).
>
> Table 1. Adversarial accuracy with different trade-off mixing strengths (ViT-B/32).
>
> | Mixing Strengths | Pets | Aircraft |
> |:-:|:-:|:-:|
> |9:1|65.8|10.1|
> |8:2|66.4|10.4|
> |7:3|64.2|10.1|
>
> **W2 & W3: Experiments on ImageNet-OOD**
>
> First, following the experimental setting used by RTPT and MTA on the ImageNet-OOD datasets, we use ResNet50 as the backbone and evaluate robustness under PGD with $\epsilon$ = 1/255 and 7 steps. As shown in Table 2, although ResNet50 only provides pooled attention from the final layer, it is still compatible with our method and achieves stronger robustness.
>
> Table 2. Adversarial accuracy on ImageNet-A and ImageNet-V2 (ResNet50).
>
> |Methods|ImageNet-A|ImageNet-V2|
> |:-:|:-:|:-:|
> |RTPT|14.4|41.6|
> |A-TPT|16.1|44.2|
>
> Second, since AOM's code isn't fully available, we couldn't exactly match its experimental setting. In addition, we compare A-TPT with a strong test-time defense method, TAPT [1], under the same AutoAttack setting. As shown in Table 3, A-TPT remains robust even under the stronger evaluation.
>
> Table 3. Adversarial accuracy compared with TAPT under AutoAttack (ViT-B/32).
>
> |Methods|Pets|Aircraft|
> |:-:|:-:|:-:|
> |TAPT|68.4|5.0|
> |A-TPT|71.2|13.8|
>
> **W5: Inference Efficiency**
>
> We further study the inference efficiency, finding that the additional computational cost stems from per-sample attention extraction, as shown in Table 4. Moreover, the inference can be accelerated by reducing the number of views to obtain better real-time performance. By reducing this number from 64 to 16, the time required for per-sample is reduced to less than 1/5 of the original, but the accuracy remains stable.
>
> Table 4. Inference efficiency on the Pets dataset (ViT-B/16).
>
> | Views Number | Time/per-sample | Accuracy |
> |:-:|:-:|:-:|
> |64|2.71 s|70.5|
> |32|1.07 s|70.1|
> |16|0.53 s|69.8|
>
> **W4 & Limitations**
>
> For multi-modal attacks, such as Co-Attack and VLATTACK, we believe that a more fundamental challenge lies in addressing the intrinsic modality gap in VLMs, rather than performing optimization solely on the image side. This limits our method in directly handling more complex multi-modal attacks on LVLMs, and it is precisely the direction we are currently pursuing in our ongoing work, as stated in Sec. 5 of the manuscript.
>
> **Q1: Symbol $t$**
>
> Thank you for pointing this out. The $t$ in Eq. (3) denotes the prompt tuning step, and we will replace it with $T-1$ in the final version.
>
> **Q2: Identifying Fine-grained Parts**
>
> Thank you for this question. Although perturbations affect all pixels, they don’t equally destroy the patch-level semantic structure of fine-grained parts.
>
> Adversarial attacks aim to maximize classification loss under a norm constraint with minimal perturbation, and therefore tend to exploit directions most sensitive to the current logit rather than destroy the multi-patch object-part scaffold. This is why our refinement lifts attention gradients to the token level.
>
> In Eq. (7), we aggregate over the token embedding dimension and use the resulting token-level weights for column scaling, making the refinement depend on the overall contribution of each source token. As a result, the final attention highlights semantic structures that still consistently respond to the target object's parts, rather than perturbed pixels.
>
> **Q3: Different Reported Results**
>
> Our results are based on the official RTPT code base, and the results of the other baselines we report on the Aircraft dataset are consistent with their originally reported results.
>
> **Q4: Clarifying Zero-shot Setting**
>
> Thank you for the suggestion, and we will clarify this in the caption in the final version.
>
> **Reference**
>
> [1] Wang, X., Chen, K., Zhang, J., Chen, J., and Ma, X. TAPT: Test-Time Adversarial Prompt Tuning for Robust Inference in Vision-Language Models. In Proc. IEEE Conf. Comp. Vis. Patt. Recogn., pp. 19910–19920, 2025a.

---

> > ### Author Rebuttal · Reviewer_37JE · 2026-04-03
> >
> > I appreciate the authors’ efforts in addressing part of my concerns in the rebuttal. I still have several follow-up questions and suggestions:
> >
> > 1. Regarding W1:
> >
> > (1) Figure 5 analyzes the number of views $N$ only at 16, 32, and 64 without baselines and the $N$ is finally set to 64. This makes it difficult to support the claim that "*A-TPT remains stable across different view numbers, indicating that only a few views are sufficient.*" Could the authors provide a comparison with the baseline under fewer views (e.g., 2/4/8) to better validate this claim? Also, I am confused what is $K$ in Line 746. Does it mean $N$?
> >
> > (2) In rebuttal Table 1, the mixing strengths are presented in ratio form. Does 8:2 correspond to weights 0.8 and 0.2? Please clarify this explicitly. Also, why does changing the mixing strength lead to noticeably different sensitivity across datasets? For example, on Pets the variation is about 2.2%, while on Aircraft it is only about 0.3%. Is a similar pattern observed on other datasets?
> >
> > (3) Could the authors further explain the decision rule for choosing the mask ratio r=0.2?
> >
> > 2. Regarding W2 & W3
> >
> > （1）Could the authors further clarify how Attention Refinement is implemented on ResNet50, given that ResNet50 does not contain attention or [CLS] token?
> > （2）I still encourage the authors to include a more complete comparison on the complete ImageNet-OOD benchmarks in the revised manuscript, similar to the evaluation protocol used in R-TPT. Also, for AutoAttack, I hope the authors can report results on a more complete set of datasets rather than only Pets and Aircraft in the revised manuscript.
> >
> > 3. Regarding W4
> >
> > The discussion on adaptive attacks is missing. I am still concerned that an adversary could exploit the proposed attention refinement/attention-guided strategy to design a stronger attack tailored to this defense, under which the method may fail. I would appreciate discussion or experiments evaluating robustness under such adaptive attack settings.
> >
> > 4. Regarding W5
> >
> > Could authors compare inference cost with baselines?
> >
> > I suggest the authors further check for typos in the revised manuscript. For example, does “mask ration” in Line 745 mean “mask ratio”?

---

> > > ### Author Response · Authors · 2026-04-05
> > >
> > > Thanks for reviewer’s detailed reply. We would like to solve the remaining concerns and misunderstandings.
> > >
> > > **W1 Response**
> > >
> > > (1) A-TPT does not use all augmented views: following the low-entropy selection step, only the top 10% low-entropy views are retained for the subsequent prompt tuning and final inference. Thus, when $N$ = 16, only 2 views remain, which is already the minimum setting that still supports genuine multi-view prompt tuning and ensemble. In contrast, when $N$ = 2/4/8, only 1 view remains, reducing the pipeline to essentially single-view inference rather than the full A-TPT framework.
> > >
> > > (2) Yes, 8:2 corresponds to $(m_{high}, m_{low})=(0.8, 0.2)$. The different sensitivity across datasets is a similar pattern on other datasets as well. However, this behavior is more consistent with datasets-specific characteristics, rather than random sensitivity.
> > >
> > > We firstly compare adversarial accuracy reduction and relative TV increase from the $(m_{high}=0.8, m_{low}=0.2)$ to $(m_{high}=0.7, m_{low}=0.3)$. As described in Eq. (12) of the manuscript, lower TV means less fragmented noises.
> > >
> > > As shown in Table 1, across datasets, the accuracy reduction increases with the degree of interference from semantically irrelevant information.
> > >
> > > As we previously stated in **Q2: Identifying Fine-grained Parts**, our refined token-gradient attention is able to recognize the correlated semantic structures on the base view that are not disrupted by adversarial noise, rather than eliminating the noise. The mixing strengths simultaneously controls the preservation of semantic information and adversarial noises on the base views.
> > >
> > >  Table 1. Adversarial accuracy reduction and relative TV increase from the $(m_{high}=0.8, m_{low}=0.2)$ to $(m_{high}=0.7, m_{low}=0.3)$ (ViT-B/32).
> > >
> > > |Changes (%)|Aircraft|Cars|Flower102|Pets|Caltech101|
> > > |:-:|:-:|:-:|:-:|:-:|:-:|
> > > |Accuracy Reduction|0.32|0.82|1.67|2.18|2.21|
> > > |Relative TV Increase|2.12|3.71|6.68|7.23|7.31|
> > >
> > > (3) As explained in Sec. C of the manuscript, we selected $r$=0.2 once via hyperparameter search on the ImageNet validation split and then fixed it across all datasets. When $r$ is too small, the protected region may miss discriminative parts; when $r$ is too large, the method over-preserves noisy regions.
> > >
> > > **W2 & W3 Response**
> > >
> > > In our ResNet50 experiment, we use the final attention-pooling layer as the only available attention source.The pooled global query in this layer serves as the counterpart of the global token, and its attention over spatial locations is used as $A(x)$. Moreover, the token-gradient weighting is retained only at the final attention-pooling layer, while the multi-layer rollout and last-2 averaging used in ViT are not applicable.
> > >
> > > Due to limited character and time, the complete results will be included in the final version.
> > >
> > > **W4 Response**
> > >
> > > We implement EOT[1] combined with BPDA[2] to approximate gradients through differentiable A-TPT’s pipeline. Specifically, we apply EOT over the stochastic multi-view pipeline by $\\nabla_x \\,\\mathbb{E}_{t \\sim \\mathcal{T}} \\big[ \\mathcal{L}(f(\\tilde{x}(A(x)))) \\big]$. This enforces the perturbation to simultaneously disrupt refined attention $A(x)$ across views.
> > > And for the non-differentiable hard masking step $M = \\mathbf{1}[A \\geq \\tau]$, we adopt BPDA with a straight-through estimator $\\frac{\\partial M}{\\partial A} \\approx 1$.
> > >
> > > Under a PGD attack ($\epsilon$ = 4/255, 7 steps) with a ViT-B/16 backbone on the Pets dataset, the accuracy of A-TPT decreases from 70.5% to 62.1%.
> > >
> > > As shown in Eq. (26) of the manuscript, our column-scaling operation $A^{(b)}(x)\\mathrm{diag}(W^{(b)}(x))$ has an additive decomposition, which eliminates the coupling between the attention matrix and the gradient of the loss with respect to the same matrix. As a result, the refined attention largely remains semantic recognition.
> > >
> > > **W5 Response**
> > >
> > > Compared with previous studies such as RTPT whose adversarial accuracy is 60.2% with running time of 0.58s/per-sample, A-TPT exhibits strong accuracy–efficiency trade-off ( The third row of Table 4 in the first rebuttal).
> > >
> > > Notably, our additional computational cost stems from per-view attention extraction and inference accuracy is stable with reducing numbers of view, which gives A-TPT a scalable accuracy-efficiency trade-off.
> > >
> > > **Typos**
> > >
> > > Thanks for your correction. We will further check for typos in the final version, such as $K$ & $N$ and ration & ratio.
> > >
> > > **References**
> > >
> > > [1] Xie, C., Wang, J., Zhang, Z., Ren, Z., and Yuille, A. Mitigating Adversarial Effects through Randomization. *arXiv:1711.01991*, 2017.
> > >
> > > [2] Athalye, A., Carlini, N., and Wagner, D. Obfuscated Gradients Give a False Sense of Security: Circumventing Defenses to Adversarial Examples. In *Proc. Int. Conf. Mach. Learn.*, pp. 274–283, 2018.

---

### Decision · Program_Chairs · 2026-04-30

**Decision:**

Accept (regular)

**Comment:**

This paper proposes A-TPT, a method for test-time adversarial adaptation of VLMs. The authors did a good job during the rebuttal, and all reviewers ultimately recommended acceptance. Thus, it is easy for the AC to recommend acceptance. However, the authors are strongly encouraged to include the additional experiments and the discussions in the final version.